# A bioactivated in vivo assembly nanotechnology fabricated NIR probe for small pancreatic tumor intraoperative imaging

Han Ren[1,8], Xiang-Zhong Zeng[1,2,3,8], Xiao-Xiao Zhao[1,8], Da-yong Hou[1,4,5], Haodong Yao [6], Muhammad Yaseen[7], Lina Zhao [6], Wan-hai Xu[4,5], Hao Wang[1] & Li-Li Li [1✉]

Real-time imaging of the tumour boundary is important during surgery to ensure that sufficient tumour tissue has been removed. However, the current fluorescence probes for bioimaging suffer from poor tumour specificity and narrow application of the imaging window used. Here, we report a bioactivated in vivo assembly (BIVA) nanotechnology, demonstrating a general optical probe with enhanced tumour accumulation and prolonged imaging window. The BIVA probe exhibits active targeting and assembly induced retention effect, which improves selectivity to tumours. The surface specific nanofiber assembly on the tumour surface increases the accumulation of probe at the boundary of the tumor. The blood circulation time of the BIVA probe is prolonged by 110 min compared to idocyanine green. The assembly induced metabolic stability broaden the difference between the tumor and background, obtaining a delayed imaging window between 8–96 h with better signal-to-background contrast (>9 folds). The fabricated BIVA probe permits precise imaging of small sized (<2 mm) orthotopic pancreatic tumors in vivo. The high specificity and sensitivity of the BIVA probe may further benefit the intraoperative imaging in a clinical setting.

[1] CAS Center for Excellence in Nanoscience, CAS Key Laboratory for Biological Effects of Nanomaterials and Nanosafety, National Center for Nanoscience and Technology (NCNST), Beijing 100190, China. [2] Center of Materials Science and Optoelectronics Engineering, University of Chinese Academy of Sciences (UCAS), 100049 Beijing, China. [3] Academy for Advanced Interdisciplinary Studies, Peking University, 100871 Beijing, China. [4] Department of Urology, The Fourth Hospital of Harbin Medical University, Heilongjiang Key Laboratory of Scientific Research in Urology, 150001 Harbin, China. [5] NHC Key Laboratory of Molecular Probes and Targeted Diagnosis and Therapy, Harbin Medical University, Harbin 150001, China. [6] Institute of High Energy Physics, Chinese Academy of Sciences (CAS), 100049 Beijing, China. [7] Institute of Chemical Sciences, University of Peshawar, Peshawar 25120, Pakistan. [8] These authors contributed equally: Han Ren, Xiang-Zhong Zeng, Xiao-Xiao Zhao. ✉email: lill@nanoctr.cn

Surgical resection remains the mainstay of treatment for patients with tumor of any grade, including pancreatic cancer, which is known as the "king of cancer"[1,2]. However, for small lesions or small metastases in pancreatic cancer (tumor diameter <2.0 cm), and the tumor boundaries, which could be hardly identified currently, are truly dependent on the experience and expertise of the surgeon[3]. Otherwise, according to the diagnosis of the cutting edge made by intraoperative frozen pathological section, the accuracy rate is only about 50%, which dramatically increased the risk of recurrence and metastasis[4]. It can be found that pancreatic cancer, which is not yet metastasized and whose tumor diameter is less than 2.0 cm will increase the 5-year survival rate to 19-41% after surgical resection[5]. Meantime, the risk of perioperative morbidity can be minimized through the combined use of MIR/CT imaging before operation and fluorescence-based intraoperation imaging[6–8].

As a real-time tumor imaging molecule, fluorescent probe has been widely used in clinical practice to provide information on tumor diagnosis and drug development[9–12]. The most famous probe is indocyanine green (ICG), which is a Food and Drug Administration (FDA)–approved near-infrared fluorescent dye, has been used in clinical intraoperative navigation[13]. These small molecules lack of active targeting and showing poor retention in tumor, which narrowed the detection window for complicated surgery[11]. For a better specificity, researchers developed tumor microenvironment turn-on probe to response ROS, lack of oxygen, pH etc[14–16]. However, traditional small molecule fluorescent dyes metabolize quickly and easily cleared by the liver and kidney, which shorted the imaging detection time[17]. Secondly, the aggregation-caused quenching (ACQ) effect of such fluorescence dyes limited the molecular accumulation in tumor and the photobleaching behavior also confused the imaging. Another way in recent years, the delivery of fluorescent probes in vivo has attracted much attention[18]. Like drug delivery[19], fluorescent probes need to be transmitted through blood to the tumor site for better osmotic enrichment at the tumor site to generate fluorescence in a stable and long-term manner[20,21]. In our previous work, we have demonstrated that the enzyme triggered in situ nanofibers assembly exhibited typical aggregation/assembly induced retention (AIR) effect to allow the fluorescent probes to accumulate near the tumor for a long time, extending the imaging time[20]. However, the imaging specificity still a great concern for challenging the small sized orthotopic tumor.

Herein, we reported a bioactivated in vivo assembly (BIVA) probe (named **M1**) for specific and sensitive pancreatic tumor imaging (Fig. 1). The BIVA probe was modular designed with five modules, including long-term circulation motif ($mPEG_{2000}$), response tailoring motif (Gly-Pro-Ala)[22], self-assembly motif (Lys-Leu-Val-Phe-Phe-Gly-Cys-Gly), targeting motif (Arg-Gly-Asp), and imaging motif ($IR_{783}$). Firstly, the probe enabled long-term blood circulation consequently enhanced the probability to reach its target[23]. Next, the targeting motif (RGD) specifically recognized and bonded onto the receptor αvβ3 integrin[24]. The over expressed fibroblast activation protein-α (FAP-α) on the membrane of MIA PaCa-2 tailored the molecules to induce in situ assembly with typical β-sheet structure[25,26]. Finally, the assembled nanofibers located around the membrane of MIA PaCa-2 in the tumor microenvironment and well labeled the tumor. Based on in vivo self-assembly nanotechnology, the BIVA probe gave optimized biodistribution half-life (long blood circulation) and elimination half-life (AIR effect), resulting in significant enhancement of area under the curve (AUC), thus increasing drug availability. In addition, the targeting and tailoring induced assembly on the membrane of MIA PaCa-2 had a spatial controlled specificity, which also contributed to enhance tumor imaging sensitivity. The introduction of mPEG module

reduces the capacity of molecular assembly of **M1** from the spatial structure and optimizes the molecular properties. Compared with previous work, we provided an optimized imaging probe delivery system based on BIVA effect, which exhibited a synergetic and enhanced new targeting mechanism: active targeting plus AIR effect[27]. We believed that such a BIVA effect and modular designed BIVA probe will offer us a tool for different imaging molecule delivery. It also has the potential to be used in intraoperative navigation.

## Results

**Molecular design, assembled behavior and conformation**. In order to study the molecular design and their assembly behavior, we modular designed and synthesized seven molecules (Table 1), which respectively were BIVA probe: **M1** (mPEG-GPAKLVFFG-C(IR783)GRGD); the non- FAP-α tailoring molecule: **M2** (mPEG-AGGKLVFFGC(IR783)GRGD) scrambled response tailoring motif of Gly-Pro-Ala with Ala-Gly-Gly; the non-labeling molecule of **M1**: **M3** (mPEG-GPAKLVFFGCGRGD) removed $IR_{783}$ labeling; FAP-α tailoring residue of **M1**: **R-M1** (AKLVFFGC(IR783) GRGD); the non-targeting molecule: **M4** (mPEG-GPAKLVFFG-C(IR783)GDTG) scrambled targeting tailoring motif of Arg-Gly-Asp with Asp-Thr-Gly; the non-labeling molecule of **M2**: **M5** (mPEG-AGGKLVFFGCGRGD) removed $IR_{783}$ labeling; FAP-α tailoring residue of **M3**: **R-M3** (AKLVFFGCGRGD). All the synthesized procedure (Supplementary Fig. 1) and characterizations of these seven molecules can be found in SI (Supplementary Figs. 2–8). As seen in Table 1, although scrambled the tailoring motif of **M2** and targeting motif of **M4**, the critical assembly concentration (CAC) of these two molecules in aqueous solution was like that of **M1**, both of which were above 500 μM. Without $IR_{783}$ labeling, molecules of **M3** and **M5** decreased the CAC value in solution relative to **M1** and **M2**, which could be attributed to the proximity of the steric hindrance of hydrophilic $IR_{783}$ to the self-assembly motif (Supplementary Fig. 9). When we removed the long-term circulation motif of $mPEG_{2000}$ tail of **M1** and **M3**, the CAC of the truncated residues of **R-M1** and **R-M3** dramatically decreased by more than two orders of magnitude (Supplementary Fig. 10). The CAC was quantitatively calculated by fluorescence probe of pyrene. All the results indicated that both hydrophilic mPEG tail and $IR_{783}$ labeling contributed to the solubility of molecules in aqueous solution.

From the molecular dynamics (MD) simulation calculations, we observed that both the backbones of **M1** and the residual **R-M1** of **M1** were β-hairpin conformations (Fig. 2 and Supplementary Fig. 11). The mPEG tail was close to the self-assembly motif through multiple hydrophobic interactions to stabilize its conformation, including hydrogen bonds VAL4:CYS8, ARG10:ALA1, ARG10: $IR_{783}$, ARG10:ASP12, ARG10:ASP12, and salt-bridge ARG10:ASP12 on both sides of the hairpin. Interestingly, the labeling of $IR_{783}$ was perpendicular to the β-hairpin backbone and mPEG tail, which formed a significant steric hindrance preventing the further intermolecular assembly. When the mPEG motif was tailored, the backbone of **R-M1** remained its β-hairpin structures by hydrophobic interactions, hydrogen bonds GLY9:PHE6, LYS2:CASP12, ARG10:CASP12, LYS2:CASP12, ARG10:CASP12 and salt-bridges LYS2:CASP12, ARG10:CASP12 on both sides of hairpin, while the $IR_{783}$ showed an obvious intramolecular rearrangement, resulting in its alignment parallel to the backbone. When the mPEG motif was tailored[28], the $IR_{783}$ was arranged parallel to the backbone, the hydrophilicity of molecule decreased and the hydrogen bonds on the self-assembled surface were exposed. The decrease of hydrophilicity of molecule and the exposure of hydrogen bonding surface of self-assembled motif were conducive for the occurrence of intermolecular dynamic

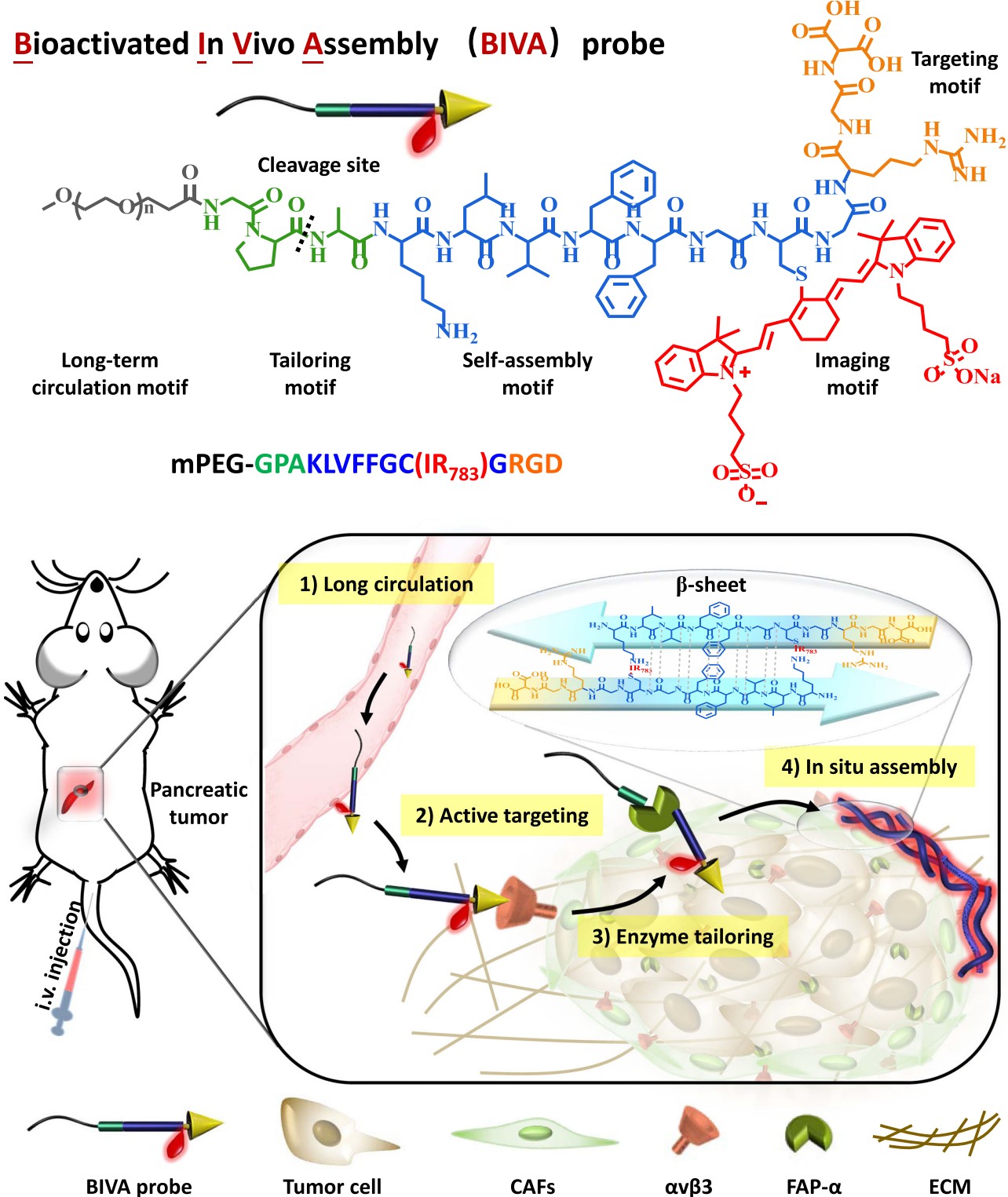

**Fig. 1 Chemical structure of bioactivated in vivo assembly (BIVA) probe.** The long-term circulation property of BIVA probe offered more opportunity for active targeting. Then, the enzyme (FAP-α) on the surface of tumor cell specifically tailored the probe and induced in situ nano-assemblies. The β-sheet assemblies exhibited a site-specific location around the tumor cell and a prolonged accumulation for better pancreatic tumor intraoperative navigation.

assembly. So that the tailoring of the mPEG motif promotes the occurrence of intermolecular dynamic assembly.

To further evaluate the assembled structures, the corresponding circular dichroism (CD), Fourier transform infrared (FTIR) spectroscopy, and wide-angle X-ray scattering (WAXS) spectroscopy were applied. As shown in Fig. 3a and Supplementary

Table 1, CD spectrum of **M3** assemblies had a positive band at λ = 193 nm and two negative bands at λ = 208 nm, and λ = 225 nm respectively, which implied a β-sheet and α-helix hybrid structure. In contrast, under the same concentration, **M1** molecules had a random coil secondary structure in CD spectrum as monomers which the concentration is lower than CAC. The

**Table 1 The sequence, assembled behavior, targeting and tailoring capability of different designed molecules.**

| Name | Molecular sequence[a] | Critical assembly concentration (CAC)[b] ($\mu$M) | Targeting receptor | Tailoring enzyme |
|------|----------------------|--------------------------------------------------|--------------------|------------------|
| M1 | mPEG-GPAKLVFFGC(IR$_{783}$)GRGD | >500 | $\alpha$v$\beta$3 | FAP-$\alpha$ |
| M2 | mPEG-AGGKLVFFGC(IR$_{783}$)GRGD | >500 | $\alpha$v$\beta$3 | – |
| M3 | mPEG-GPAKLVFFGCGRGD | >303.7 | $\alpha$v$\beta$3 | FAP-$\alpha$ |
| R-M1 | AKLVFFGC(IR$_{783}$)GRGD | 37.2 | $\alpha$v$\beta$3 | – |
| M4 | mPEG-GPAKLVFFGC(IR$_{783}$)GDTG | >500 | – | FAP-$\alpha$ |
| M5 | mPEG-AGGKLVFFGCGRGD | >332.7 | $\alpha$v$\beta$3 | – |
| R-M3 | AKLVFFGCGRGD | 19.2 | $\alpha$v$\beta$3 | – |

[a]mPEG was methoxy polyethylene glycol with a molecular weight around 2000.
[b]The CAC was analyzed based on pyrene.

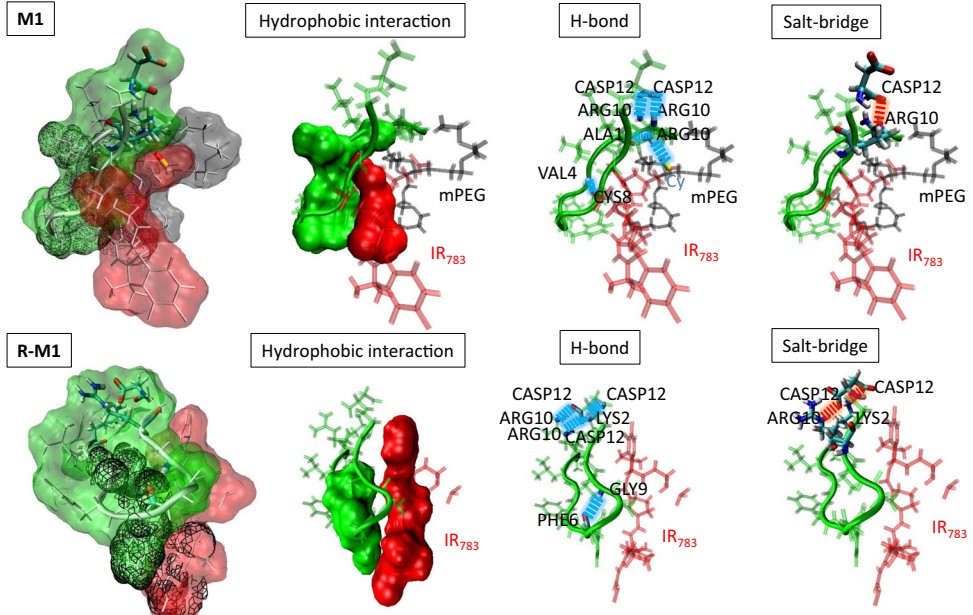

**Fig. 2 The molecular dynamics (MD) simulation of M1 and R-M1.** The molecular interaction details including hydrogen bond (blue dotted-line), salt-bridge (red dotted-line) and hydrophobic interaction. Red part: IR$_{783}$; Gray part: mPEG$_{2000}$; Green part: peptide.

FITR spectra of **M1** and **M3** in Fig. 3b observed the intermolecular interactions. The peaks at 1629 cm$^{-1}$, 1675 cm$^{-1}$, and 1698 cm$^{-1}$ of **M3** indicated anti-parallel $\beta$-sheet structure (represent by green arrow heads), peaks at 1648 cm$^{-1}$ and 1663 cm$^{-1}$ indicated parallel beta sheet structure (represent by blue arrow heads), the existence of 1654 cm$^{-1}$ indicate $\alpha$-Helix structure in **M1** which verified the results deduced by CD in Fig. 3a[29,30]. The evidence indicated that without IR$_{783}$ labeling, the **M3** molecule was easier to assemble than the **M1** in the absence of IR$_{783}$, and the driving forces of assembly depend on the multiple hydrogen bonds and other weak interactions of the self-assembly motif. After tailoring the hydrophilic balance of mPEG, the **R-M1** exhibited a well-ordered $\beta$-sheet assembled secondary structures with a typical strong positive band at 196 nm and a wide negative band at 216 nm (Fig. 3c). The **R-M1** molecules had a rapid dynamic assembly (within few minutes), and the assemblies in aqueous solution had an obvious Tyndall phenomenon. As a homologous sequence with amyloid $\beta$-protein (A$\beta$), the self-assembly motif with peptide sequence of KLVFFGCG had similar aggregation kinetics to (A$\beta$)42 peptide, which occurred via dynamic growth from oligomers to amyloid fibrils[31]. The aggregation started from the freshly isolated monomers of **R-M1**, and precipitates were separated in 1 min and 1 h, respectively. The FTIR spectra of these two samples (Fig. 3d) exhibited completely different spectral features. The one

separated rapidly showed a broad peak at 1634 cm$^{-1}$, which was identified as oligomer; while the one with extended aggregated time had three peaks at 1698 cm$^{-1}$, 1688 cm$^{-1}$, and 1629 cm$^{-1}$, which were ascribed to the as anti-parallel $\beta$-sheet fibrils[32,33]. After analyzed the CD spectra of **M1**, **M3**, and **R-M1** (Fig. 3e), it can be clearly seen that the main secondary conformation of **M1** was Random, the **M3** was hybrid of Helix and Beta, and the **R-M1** was Beta (Supplementary Table 1). All the results confirmed the conclusion from FTIR spectra. Characteristic of nucleated growth procedure (Fig. 3f), the aggregation curves with ThT trace had a growth phase for primary process from the initial 17 min, an elongation phase for surface-catalyzed secondary process between 17 and 30 min, and a final plain phase after 30 min. The dynamic growth procedure was like the 8-anilino-1-naphthalenesulfonic acid (ANS) stained curve (Supplementary Fig. 12). As known, ANS was sensitive to hydrophobic interaction[34]. When **R-M1** were in the initial oligomer, the fluorescence intensity of ANS increased due to the enhanced hydrophobicity. When the molecules were elongated and stacked in higher ordered nanofibrils, the blue shift of ANS in the $\beta$-sheet structures reduced the fluorescence. The molecular packing mode of well-ordered fibrils of **R-M1** in Fig. 3g observed a weak reflection at 4.9 Å as laminates space and a strong broad reflection at 10.3 Å as sheet space, which was illustrated in the inserted figure. The fibril morphology was characterized by transmission electron

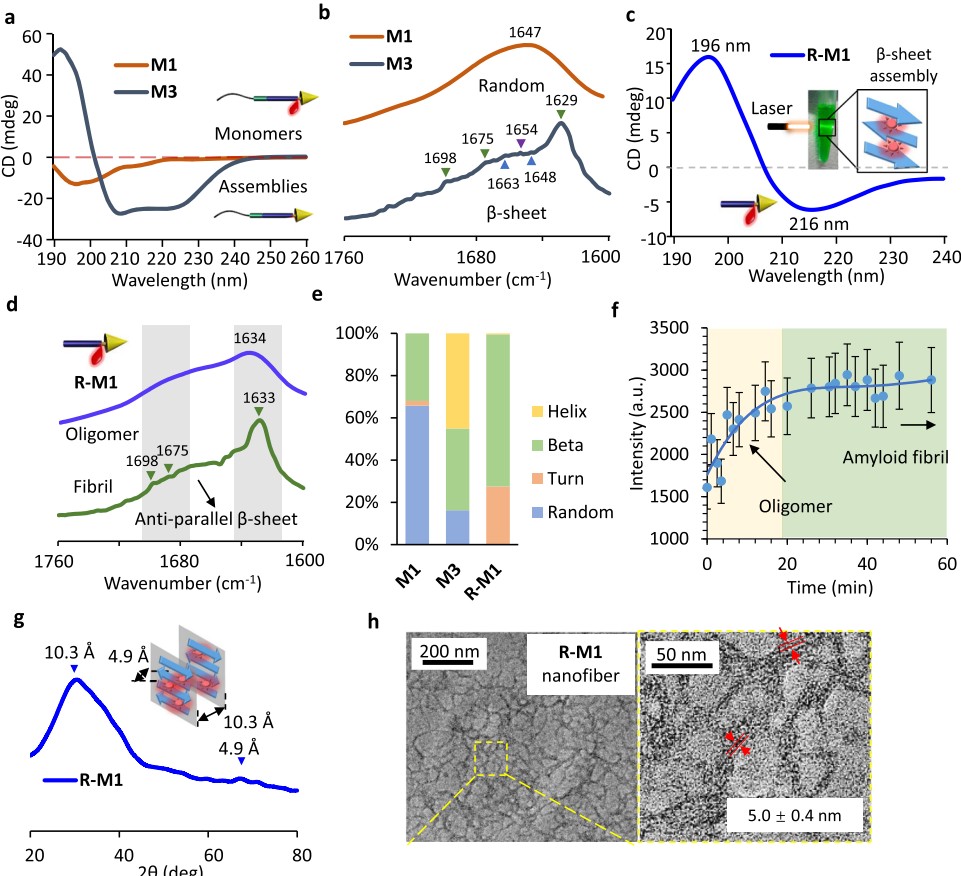

**Fig. 3 Assembled structure conformations and self-assembly behavior in aqueous solution. a** CD spectra of **M1** and **M3** in DI water under a concentration of 200 μM. **b** The FTIR spectra of **M1** and **M3**, powder samples collected from freeze dried sample solutions. **c** The typical β-sheet CD spectra of **R-M1** in DI water (insert figure: the Tyndall phenomenon) under a concentration of 100 μM. **d** The FTIR spectra of dynamic growth of **R-M1**, powder samples collected from freeze dried sample solutions in different period. **e** Analysis of secondary structure composition of **M1**, **M3**, and **R-M1** based on CD spectra. **f**, Elongation-nucleation growth procedure with ThT staining. The mean of data of three samples with the same conditions is shown and data are presented as mean values ± SD (*n* = 3) **g**, The WAXS spectrum and illustration of the **R-M1** fibrils, powder sample collected from freeze dried sample solution. **h**, The TEM images of nanofibers morphology of **R-M1**.

microscopy (TEM) imaging (Fig. 3f). The statistical calculation of the fiber diameter in TEM images was 5.0 ± 0.4 nm (Supplementary Fig. 13), which was corresponded to the theoretical calculated two molecules length of **R-M1**. We assumed that the nanofibers were assembled by twisted **R-M1** molecules centered on self-assembly motif.

**Specific enzyme tailoring induced nanofibril assembly**. To further investigate the FAP-α specific tailoring and BIVA probe assembly in situ simultaneously (Fig. 4a), the high-performance liquid chromatography (HPLC), TEM, and matrix-assisted laser desorption/ionization time-of-flight (MALDI-TOF) mass spectrometry were applied for the characterizations purpose. To specify the cutting of the response tailoring motif (Gly-Pro-Ala), pre-synthesized molecules of **R-M3**, **M3** and **M5** were set as controls. After incubation with FAP-α for 12 h, **M3** molecules were totally cleaved by the enzyme (Fig. 4b), resulting in truncated residues with different retention times compared to **M3** (29.5 min). Comparing the residues peaks with **R-M3**, the primary sharp peak at 27.4 min can be identified by the **R-M3** control (27.6 min). In the meantime, the wide peak at 14.6 min might be the remaining PEG residue. In sharp contrast, after incubation of FAP-α with **M5**, there was no change of the retention peak, which double confirmed that the GPA was the FAP-α specific recognized sequence and the molecule was cut

between the amino acid of Pro and Ala. The retention time of **M1** after co-incubation with inactivated FAP-α was similar to that of **M3** (Supplementary Fig. 14). After hydrolyzed by FAP-α, the residues of **M1** in situ assembled into nanofibers structures under the complex buffer (Fig. 4c and Supplementary Fig. 15). **M1** could not be assembled after co-incubation with the inactivated FAP-α, which further indicated that the assembly of **M1** could be ascribed to the FAP-α tailoring (Supplementary Fig. 16). The tailored residues were identified by MALDI-TOF (Fig. 4d), which revealed that **M1** was cut into two parts of **R-M1** and PEG residue. Additionally, when the molecules of **M1** were incubated with MIA PaCa-2 cells for 2 h, the cell lysis was observed to split into layers. MALDI-TOF confirmed that **R-M1** in the precipitate might be the assembly induced precipitate, and the supernatant obviously contained the PEG residues (Supplementary Fig. 17). In order to further evaluate the specificity of **M1** for different enzymes, including FAP-α, pepsin, pancreatin, lipase, and BSA, we used ThT (Thioflavin T) as a detection probe (Fig. 4e). After co-incubation with **M1** for 12 h, only FAP-α group strikingly enhanced the fluorescence intensity, which indicated that **M1** molecule had specificity for FAP-α induced in situ assembly.

**Enhanced targeting and in situ high efficiency nanofiber formation located cell outline**. As designed, the bioactivated in vivo assembly (BIVA) was a triggered and synchronous dynamic

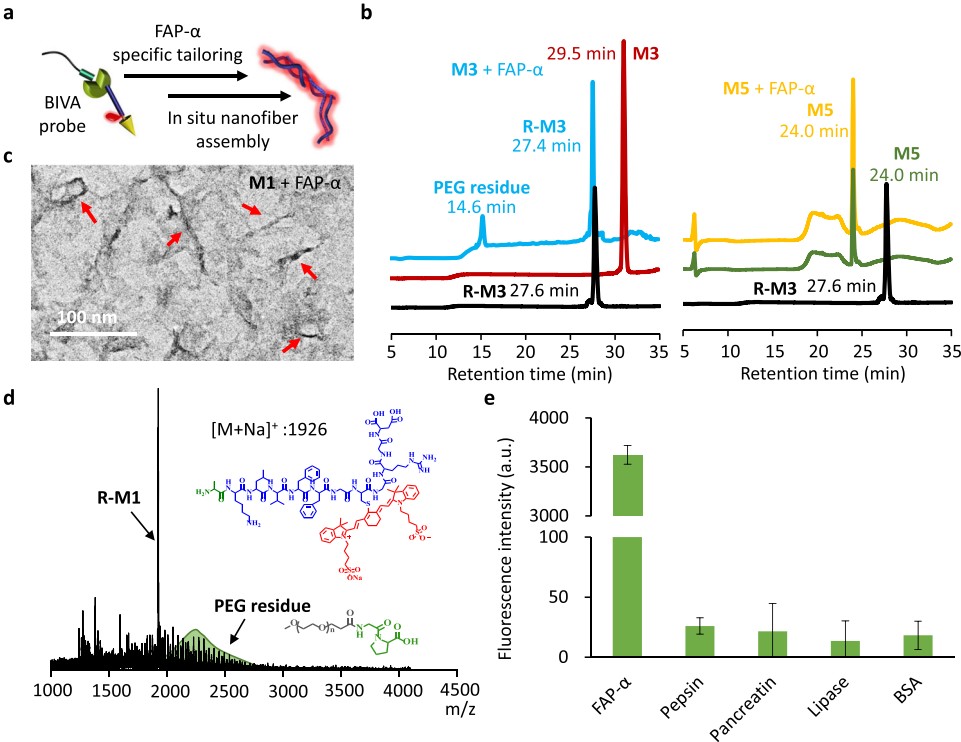

**Fig. 4 The FAP-α specific molecule tailoring and inducing in situ assembly. a** The illustration the working mechanism of BIVA probe based on FAP-α catalytic hydrolysis. **b** HPLC curves of **M3** and **M5** after incubation with FAP-α in buffer. **R-M3**, **M3** and **M5** were synthesized controls. The TEM image (**c**), and the MALDI-TOF (**d**), results of **M1** after tailoring by FAP-α in buffer, the red arrows represent the assembled fibrils of M1 after incubation with FAP-α. **e** The enzyme specificity of **M1** in buffer. Buffer: 50 mM Tris, 1 M NaCl, 1 mg/mL, BSA, pH 7.5; Concentration of **M1** and **M3**: 100 μM; Concentration of FAP-α: 50 μM; incubation time: 12 h. The mean of data of four samples with the same conditions is shown and data are presented as mean values ± SD ($n = 4$). $p = 2.04E-22 < 0.001$, $p$ values were performed with one-way ANOVA by post hoc Tukey's test for the indicated comparison.

assembly system with active targeting cooperative aggregation/assembly induced retention (AIR) effect. Compared with active targeting mechanism dependent on binding constant Kd, BIVA effect showed an amplified mechanism based on primary binding constant Kd and secondary assembly rate constant Ka (Fig. 5a). To further confirm this hypothesis, the **M1** and **M2** were equally labeled by FITC for cell imaging (Fig. 5b). To simulate the dynamic physiological condition, the cell culture medium was replaced every 15 min. After 1 h of incubation with **M1** and **M2** at the same molecule molar concentration, there occurred significant differences between the two molecules (Fig. 5c). The **M1** with BIVA effect had a higher fluorescence retention rate on the cell membrane, while the **M2** with active targeting mechanism significantly reduced the fluorescence signal during the dynamic incubation. The huge difference can be explained by the fact that the secondary assembly rate broke the balance between the targeted ligand and receptor, and thus tended to a stable assembly interaction. The retention efficiency depended on the assembly rate constant Ka. Meanwhile, the rapid dynamic assembly of BIVA probe in situ around the cell membrane, and contributed to the efficient formation and retention of nanofibers on the cell profile. When the incubation time was delayed up to 2 h, some components could be endocytosed into the cells, but most of them were found on the membranes (Supplementary Fig. 18). As shown in Fig. 5d, most molecules of **M1** were assembled and located on the cell membrane within 1 h of incubation. Then, the isolated cells were collected and lysed, the extracted cell membrane fragments were stained with ThT. Interestingly, the nanofibers on the membrane were all stained by ThT, and the correlation coefficient between ThT and FITC fluorescence was high up to 0.83 (Fig. 5e). In order to verify the influence of the

addition of the targeting module of the probe molecules, we explored the imaging of **R-M1** and **M4** on cells. After co-incubation with **R-M1**, which was an insoluble suspended mixture, the fluorescence signal could be observed both in medium and on the cell membrane (Supplementary Fig. 19d). While the cells were treated by **M4**, only a small amount of fluorescence signal could be observed on cell membrane (Supplementary Fig. 19e). The phenomenon observed by **R-M1** indicated that with RGD motif, the assembled **R-M1** still remain the specific bonding capability to cell membrane. Without RGD bonding, the **M4** molecules could also be tailored by FAP-α and the assembled nanofibers exhibited non-specific interaction on the membranes. Moreover, once pre-treated the cells with RGD peptides (Supplementary Fig. 19b) or FAP-α inhibitors (Supplementary Fig. 19e), the cell membrane bonding and retention of the **M1** signals were significantly reduced, which validated that the RGD recognition and FAP-α induced nanofibers formation both contributed to the attach on the membranes.

In order to further understand the contribution of the active targeting and AIR effect during locating of the cells, the cell transwell experiment was used to quantitively evaluate interference of the cell migration based on **M1**, **M2**, and **M4** (Fig. 5f). As expected, untreated cells were easy to migrate to the lower chamber, while the cells treated with active targeting molecule **M2** and assembled molecule **M4**, the migration of cell were reduced. Under the same molar concentration, the **M1** treatment group had the most inference on cell migration (Fig. 5g). According to the quantitative statistical calculation of the number of cells (Fig. 5h and Supplementary Fig. 20), the results clearly verified that the BIVA effect indeed had a high trapping and localization efficiency around cells. Based on those results, we speculate **M1**

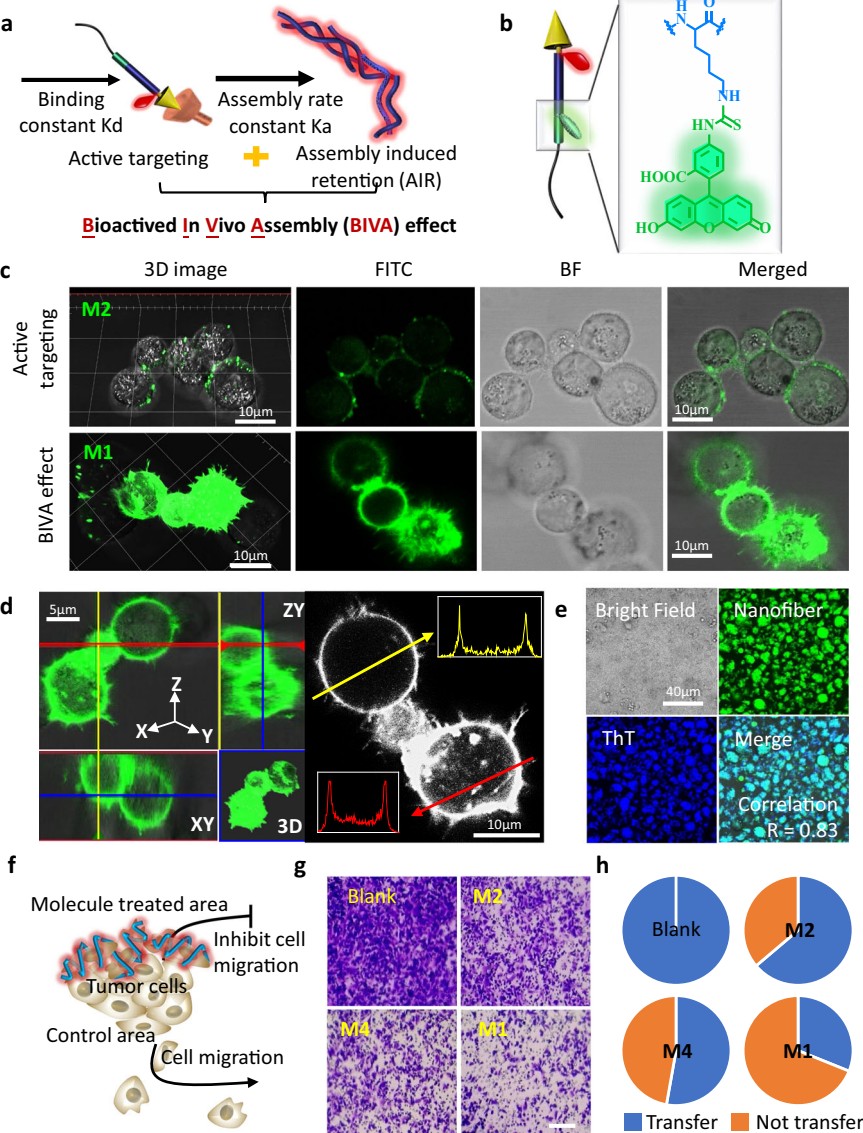

**Fig. 5 The BIVA effect with enhanced targeting to pancreatic tumor cell for boundaries imaging. a** Illustration of BIVA effect: active targeting and assembly/aggregate induced retention (AIR) effect. **b** Chemical structure of FITC labeling. **c** 2D and 3D confocal images of **M2** with active targeting property and **M1** with BIVA effect on MIA PaCa-2 cells after incubation for 1 h. **d** The distribution of **M1** on MIA PaCa-2 cell. **e** ThT staining of the lysed cell membrane of MIA PaCa-2. **f**, Illustration of migration inhibition after treated by BIVA probe. **g** The image of the migrated cells after treatment of PBS (blank), **M2**, **M4**, and **M1**. Scale bar: 100 μm. **h** The pie diagram of quantitative statistical calculation of the migrated cells (blue) in different groups. The blank control was set as 100% migrated cells.

can be more enriched in the tumor site and form fibers through assembly to inhibit tumor cell migration compared to **M2** and **M4** which have no ability to target and adhere on the cell membrane. Although the targeting group of **M2** and BIVA group of **M1** had significant interaction with cells, there was no obvious cytotoxicity at high concentration of 300 μM (Supplementary Fig. 21).

**Metabolic difference and optimized biodistribution enhanced imaging.** In order to reduce systemic error and individual difference, we constructed the subcutaneous pancreatic cancer model in mice for quantitatively calculating of the metabolic data. First, the subcutaneous tumor model of the right hind leg had no overlap with other organs, and it can reduce the organs depth difference caused system error. Secondly, compared with the orthotopic tumor, the size of the subcutaneous tumor was more controllable, reducing the individual difference between the

experimental groups, and making the experimental data more reliable. According to the time-dependent in vivo NIR images, there were significant differences in the fluorescence distribution among ICG, **M2**, and **M1** mice (Fig. 6a). The representative small molecule probe was ICG, which showed rapid distribution and elimination all over the mice body with no obviously specific targeting effect on tumor tissue. The results of near infrared (NIR) imaging showed that ICG completed its metabolic clearance within 8 h. Moreover, the **M2** probe with active targeting capability was distributed and accumulated in the tumor area within 0.1–24 h. The metabolized rate from tumor and other tissues seemed no obvious difference. However, the **M1** probe based on the BIVA effect optimized the biodistribution, which accumulated more signal in the tumor area and enhanced retention in tumor long lasted up to 96 h. Meantime, the non-tumor tissues showed lower signal distribution resulting a short elimination time. Based on NIR imaging, the **M1** with higher

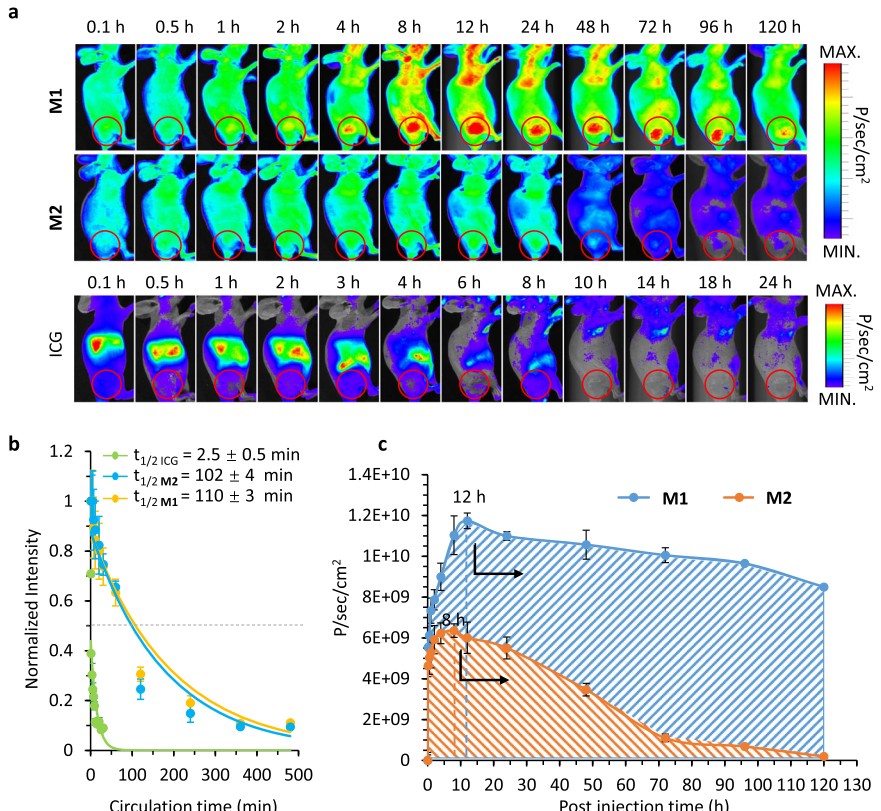

**Fig. 6 The BIVA effect optimized the metabolism of the probe in vivo. a** The time-dependent NIR fluorescence image of mice bearing MIA PaCa-2 cells after intravenous administration of ICG, **M2** and **M1** with a dose of 16 mg/kg. The images acquired at time intervals from 0.1 h to 120 h are managed with the same procedure (The circles represent the locations of the tumors). **b** The blood circulation curve of ICG, **M2** and **M1** based on exponential curve fitting. The $t_{1/2}$ value was the blood circulation half-life. **c** The time-dependent quantitative calculation of the average fluorescence intensity in tumor area and the area under the curve (AUC) of ICG, **M2** and **M1**. The mean of three biological replicates is shown and data are presented as mean values ± SD ($n = 3$).

imaging contrast enable to have a stable detection window within 8 h-96 h attributing to the enhanced targeting and metabolic difference of the BIVA probe. Meanwhile, the photoacoustic imaging after i.v. injection of **M1** and **M2** for 12 h clearly show the big difference on the tumor signal and the surrounding tissues, the **M1** had more accumulation in tumor than that of **M2** (Supplementary Fig. 22).

Otherwise, the molecules of **M1** and **M2** with a mPEG tail, both had long-time circulation half-life ($t_{1/2}$), which were 110 ± 3 min and 102 ± 4 min, respectively (Fig. 6b). The blood circulation half-life ($t_{1/2}$) of ICG was 2.5 ± 0.5 min. The short $t_{1/2}$ was related to the rapid distribution and elimination behavior in vivo. In order to understand the contribution of these elements to effective availability of imaging probe. The significant parameter of pharmacokinetic: area under the curve (AUC) in tumor tissue was obtained according to the quantitative calculation from fluorescence signal. After quantitative calculation the concentration of probe in tumor area without background signal subtracted, the time-dependent curve of **M1** and **M2** were carried out (Fig. 6c). The area under the curve ranges from 0 h to 120 h (AUC 0-120 h) of **M1** was 3.6 times more than that of **M2**, which mean that with a single dose administration, the average fluorescence intensity distribution per unit area of **M1** in tumor tissue was 3.6-fold higher than that of **M2**. The time to peak of **M1** was 4 h later than **M2**, about at 12 h. The highest signal on tumor of **M1** was 1.8 times higher than **M2**. In addition, the signal elimination of **M1** from tumor was quite slow, only 27.6% was reduced between 12 h and 120 h, while the fluorescence signal of **M2** was disappeared completely in the same time interval. Finally, we obtained a stable intraoperative navigation window between 8 h and 96 h for our

BIVA probe (**M1**). In conclusion, both the long-term blood circulation and the dynamic enzyme tailoring helped the continuous accumulation in tumor area. The in-situ assembly in tumor tissues slowed down the dynamic elimination and prolong the elimination time, which contributed to maintaining the imaging signal during surgery operation. The FAP-α specific tailoring and assembling of **M1** differed the tumor from the other tissues, which offered better contrast and biodistribution. To evaluate the imaging property of BIVA probe, the orthotopic pancreatic tumor mice model was built. After intravenous injected **M1** and **M2** molecules with a dose of 16 mg/kg for 12 h, the mice were sacrificed. When dissected the spleen, the high contrast signal was clearly observed on the orthotopic tumor area (Fig. 7a, Supplementary Fig. 23). Then, all the important organs were dissected for ex vivo imaging. The significant difference between **M1** and **M2** on the tissue biodistribution. For BIVA probe **M1**, the distribution on tumor had obviously selectivity, and the molecules had part retention in the metabolic organs (e.g., liver and kidney). Whereas, the **M2** exhibited no significant difference in the biodistribution of lung, kidney, and tumor, but most of the molecules were stuck in liver. The huge difference between the two molecules can be explained by the high specific recognition of FAP-α to **M1**, which was conducive to efficient molecular tailoring and assembly in tumor, while the **M2** was non-specific cleavage and accumulation in liver during metabolism. The quantitative analysis results also confirmed the conclusion (Fig. 7b). The signal accumulation of FAP-α specific BIVA probe **M1** was twice as much as that of **M2**. Under the same blood circulation time, organ selectivity depended on the specificity of substrate to target enzyme. Its accumulation amount relied on the

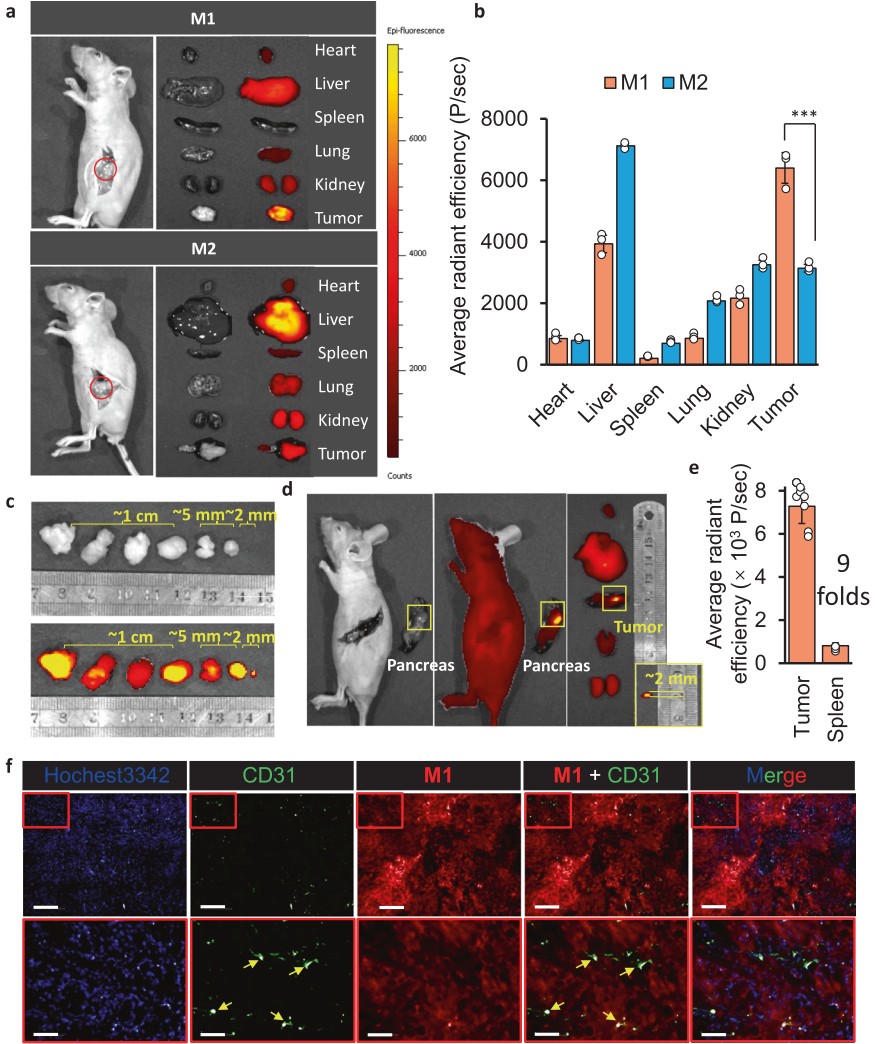

**Fig. 7 The BIVA probe enhance orthotopic pancreatic tumor imaging in vivo. a** The in vivo NIR images of small orthotopic pancreatic tumor by **M1** and **M2**, and the ex vivo of organ biodistribution including heart, liver, spleen, lung, kidney, and tumor. The mean of three biological replicates is shown ($n = 3$). **b** The quantitative analysis of average fluorescence intensity per organ area $p = 0.000844 < 0.001$, ($n = 3$). **c** The in vivo NIR images of orthotopic pancreatic tumor with individual difference with the same i.v. dose administration for 12 h. The mean of five biological replicates is shown ($n = 5$). **d** The small size (~2 mm diameter) orthotopic pancreatic tumor images and its ex vivo signal distribution. **e** The quantitative calculation of the signal in tumor area and healthy spleen area. The mean of eight biological replicates is shown. ($n = 10$) **f** The tumor histochemical staining with Hochest3342 (blue), CD31 (green) and **M1** (red) post i.v. injection of **M1** for 48 h. The yellow arrows pointed at the blood vessels. Bars of the up layer: 200 μm; Bars of the bottom layer: 50 μm. Data: mean ± standard deviation. Injection dose (i.v. administration): 16 mg/kg. Statistical analysis: one-way t test followed by post hoc Tukey's test, ***p < 0.001.

cleavage rate of enzyme and aggregation efficiency of molecular residues. The primary nucleation of assembly can induce long lasting growth of the fibril in tumor, reduce the metabolic rate, and achieve the retention and accumulation of tumor. When pre-treated the mice with **M1** and **M2** with the same dose for 48 h (Supplementary Fig. 24), the signal in liver were reduced, which could be explained as the dynamic metabolism by the liver. The signal on orthotopic tumor or small tumor still clearly observed after 48 h **M1** treatment. Upon the individual differences, we validated 6 mice under surgery to induce orthotopic tumor in pancreatic head. All the positive results were obtained including the small sized tumor around 2 mm in the diameter (Fig. 7c). The ex vivo dissection in Fig. 7d provided a fantastic imaging contrast on tumor and the around the spleen tissue, which visualized and identified the small tumor (~ 2 mm) both on 2D and 3D images (Supplementary Fig. 25). The statistical results of fluorescence signal on tumor were over 9 folds higher than the surrounding spleen tissue (Fig. 7e). The whole tumor histologic

section in Supplementary Fig. 26 stained by Congo Red fully viewed the fibril distribution the tumor after 12 h **M1** treatment. As known, the FAP-α was a membrane located protein, overexpressed on tumor associate fibroblast cell and pancreatic cell surface. The FAP-α specific BIVA probe **M1** were well depicted the tumor margin and interstitial space, which concentrated the signal in tumor for better bioimaging, but the **M2** has no obvious Congo Red, which means there was no assembly inside tumor. In order to validate the **M1** deep penetration in tumor after 48 h **M1** administration, we merged the tumor histochemical staining with Hochest3342 (blue), CD31 (green) and **M1** (red). As seen in Fig. 7f, most of the **M1** were far away from the green colored blood vessels and the signals were uniform distributed in tumor. The whole tumor section both confirmed the conclusion above (Supplementary Fig. 27). Once stained by Congo Red (Supplementary Fig. 28), the tumor slices were easily observed the red distribution, which mean that the deep penetrated **M1** were transformed to nanofibers.

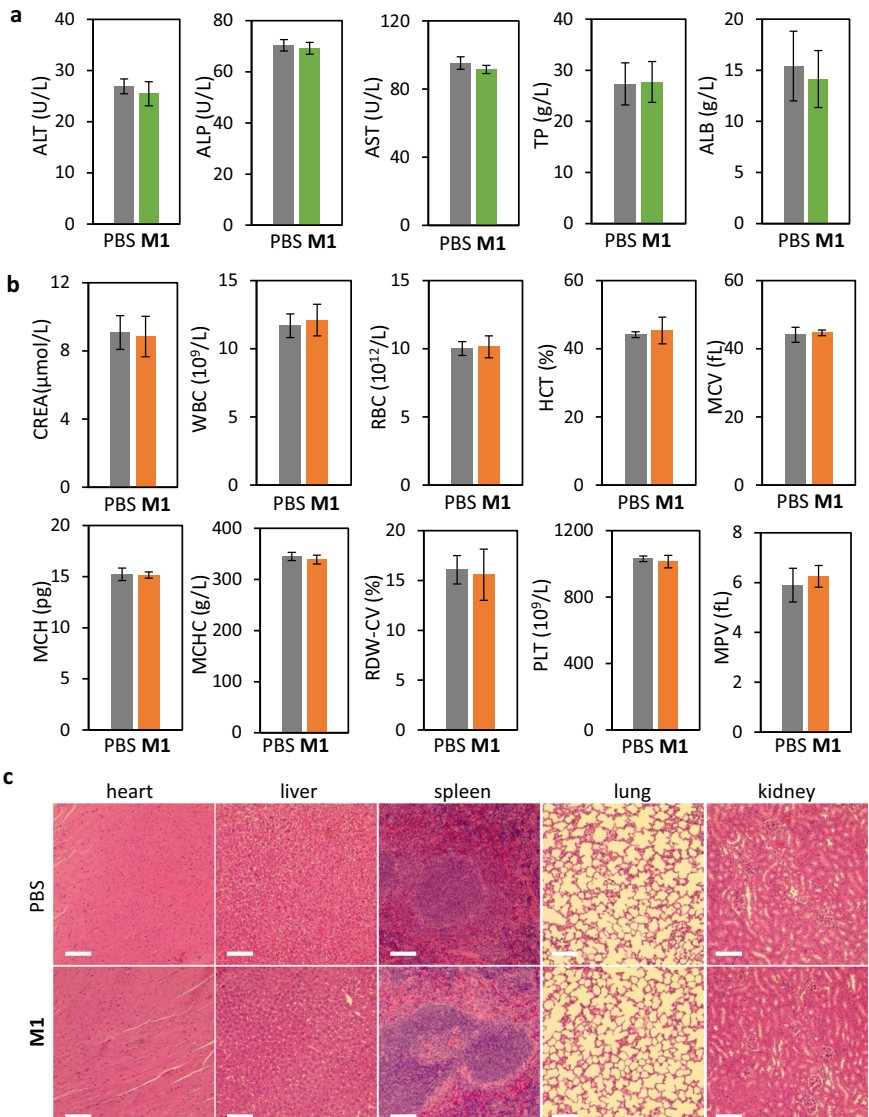

**Fig. 8 Acute toxicity evaluation. a** Liver function indicators: alanine aminotransferase (ALT), alkaline phosphatase (ALP), aspartate aminotransferase (AST), total protein (TP) and albumin concentration (ALB). The mean of four biological replicates is shown and data are presented as mean values ± SD ($n = 4$). **b** Blood biochemical indicators: creatinine (CREA), white blood cells (WBC), red blood cells (RBC), hematocrit (HCT), mean corpuscular volume (WCV), mean corpuscular haemoglobin concentration (MCH), mean corpuscular hemoglobin concentration (MCHC), red cell volume distribution width-coefficient of variation (RDW-CV), platelet (PLT) and mean platelet volume (MPV). The mean of four biological replicates is shown and data are presented as mean values ± SD ($n = 4$). **c** Histologic sections of different organs: heart, liver, spleen, lung, kidney and pancreas compared with healthy group (PBS). Staining: H&E; Injection dose of M1. (i.v. administration): 16 mg/kg; Administration time: 24 h. Scale bar 200 μm.

**Acute toxicity evaluation to organs**. The acute toxicity evaluation of **M1** to mice were verified by blood biochemistry, hemograms, and histological analysis. The representative biomarkers of liver function included alanine aminotransferase (ALT), alkaline phosphatase (ALP), aspartate aminotransferase (AST), total protein (TP), and albumin concentration (ALB). Compared to the healthy group (PBS), there was no obvious hepatic toxicity after i.v. injection of 16 mg/kg of **M1** for 24 h (Fig. 8a). In addition, the hematological assessment results including creatinine (CREA), white blood cells (WBC), red blood cells (RBC), hematocrit (HCT), mean corpuscular volume (WCV), mean corpuscular hemoglobin concentration (MCH), mean corpuscular hemoglobin concentration (MCHC), red cell volume distribution width-coefficient of variation (RDW-CV), platelet (PLT) and mean platelet volume (MPV) were carried out (Fig. 8b). All the above indicators in PBS group and **M1** group appeared normal, which was basically consistent with the normal range reported in the literature. Then the mice from PBS group and **M1** group were

sacrificed for the further histological section analysis of the significant organs. After Hematoxylin and Eosin (H&E) staining, the slices of heart, liver, spleen, lung, kidney, and pancreas were compared and evaluated (Fig. 8c). There was no noticeable organ damage and tissue injury of the two groups. All the evidence revealed that under the imaging dose, the BIVA probe had no acute toxicity performance.

## Discussion

Although the optical probes had great potential in the further clinical trials, the drug-ability still a bid issue. As a great improvement, the targeting ligand covalent coupling such as antibody, peptide and etc., significantly increased the targeting property of the probes. However, the optical probes based on active targeting mechanism were always in a narrow imaging window, which cannot meet the need of stable imaging for

intraoperative navigation. Thus, we proposed a bioactivated in vivo assembly (BIVA) nanotechnology to fabricate NIR probe, which offered a new targeting mechanism based on synergy of active targeting and AIR effect. The existence of mPEG tail of the probe stabilized the molecular conformation with better hydrophilicity for receptor recognition and long blood circulation half-life ($t_{1/2}$) up to 110 min. Then, such modular designed BIVA probe highly enriched on tumor cell membrane through a dynamic ligand-receptor binding and sequently FAP-α triggered self-assembly process, resulting in a high contrast imaging with a signal-to-background ratio SBR over 9 folds. The site-specific assembly contributed to great difference in metabolized time between tumor cells and surround healthy cells, further optimizing the organ distribution. Finally, the in situ assembled fibrils located in tumor region reduced the degradation and metabolism of the probe, which broadened the imaging window to 8 h~96 h. All the efforts realized a highly sensitive small (<2 mm) orthotopic pancreatic tumor imaging in vivo. We believed that the BIVA nanotechnology would upgrade the imaging probe into clinical drug. The dynamic assembled property and assembled conformation modulation would be optimized according to the surgical imaging needs. Such a delivery system can further be expanded to carry different fluorescent molecules, the resulted BIVA probes will benefit future precision medicine.

## Methods

**Materials**. All Fmoc protected amino acids, O-(benzotriazol-1-yl)-N, N, N', N'-tetramethyluronium hexafluorophosphate (HBTU) trifluoroacetic acid (TFA), 1,2-ethanedithiol, triisopropulsilane (TIPS), 2,5-dihydroxybenzoic acid (DHB) are bought from Sigma-Aldrich Chemical Co. Dimethyl formamide (DMF), dichloride methane (DCM), piperidine, N-methylmorpholine (NMM), diethyl ether, hexafluoroisopropanol (HFIP), dimethyl sulfoxide (DMSO), uranyl acetate, methanol, acetonitrile, diamine tetraacetic acid (EDTA-2Na) and 4% formaldehyde solution were purchased from Aladdin technology (Shanghai) Co. Wang resins were purchased from GL Biochem (Shanghai, China). Dulbecco's Modified Eagle Media (DMEM), Fetal Bovine Serum (FBS), Penicillin and streptomycin, Phosphate-Buffered Saline (PBS) and Trypsin were purchased from HyClone/Thermofisher (Beijing, China). Cell culture plates and Transwell assay kit were purchased from Coning Company. CCK-8 (Cell counting kit-8) was purchased from Beyotime Biotechnology Co., Ltd. (Shanghai, China). The MiaPaCa-2 cell lines were purchased from the cell culture center of the Institute of Basic Medical Sciences, Chinese Academy of Medical Sciences (Beijing, China). Female BALB/c mice (6–8 weeks, 16–18 g) were purchased from Vital River laboratory animal technology Co., Ltd. (Beijing, China). Recombinant Human Fibroblast Activation Protein-α (FAP-α) was purchased from R&D SYSTEM. Other solvents used in the research were purchased from commercial companies.

**Molecule synthesis and characterization**. The peptides were synthesized according to the standard solid phase synthesis method[35]. The loading value of Wang resin was 0.349 mmol/g, where the de-protective agent was a mixture of anhydrous dimethylformamide (DMF) and piperidine with a volume ratio of 4/1. The coupling agent was composed of 5% NMM and 95% anhydrous DMF (V/V). The cleavage agent was a mixture of 92.5% TFA, 2.5% TIPS, 2.5% 1,2-ethanedithiol, 2.5% $H_2O$ (V/V/V/V) solution. mPEG$_{2000}$-NHS was linked to the N terminal of peptide similar with other amino acid but the reaction time lasted up to 24 h. The IR$_{783}$ (1.1 mol equivalent) was coupled onto peptide (cleaved from the resin) after reacting in Tris buffer at pH = 8.5 for 12 h, washed the unreacted dye in ice bath with DCM, and dialyzed on MWCO 2000 Da dialysis membrane for 24 h. Finally, all the molecules were characterized by MALDI-TOF-MS (Matrix: saturated DHB in TA30) and HPLC having a C18 column and a linear gradient of acetonitrile/water with 0.1% TFA from 10%/90% to 50%/ 50% and a flow speed of 1 mL/min at 25 °C.

**MD simulation**. The GROMACS software package (version 5.1.4) was used to perform energy minimization and molecular dynamics simulations using the AMBER99SB-ILDN force field. The topology and force field parameters files for the PEG and the rest of the molecule were generated with AmberTools and Acpype Antechamber. All the structures were solvated in a box of TIP3P water model, and then ionized and neutralized with Na$^+$ and Cl$^-$ ions as 0.15 mol/L. The periodic boundary conditions (PBC) were set for all directions. The NPT ensemble was applied. After energy minimization, each structure underwent NVT equilibration and NPT production. The Nose-Hoover thermostat[36] was used to maintain the temperature of system as 300 K. The pressure was maintained as 1 atm by coupling the semi-isotropic (X + Y, Z) directions of the system using the Parrinello

−Rahman algorithm[37]. The van der Waals (vdWs) interactions were computed with a cutoff distance as 12 Å, while long-range electrostatic interactions were handled with the particle mesh Ewald (PME) method[38]. The water molecules were constrained by the SETTLE algorithm[39], and the hydrogen bonds were constrained to their equilibrium values employing the LINCS algorithm[40]. The time-step was 2 fs in production runs, and the coordinates were saved every 100 ps. Total time of each system was 200 ns. The VMD software was used to observe and draw the image of all systems.

**CD spectroscopy**. The secondary structures of **M1**, **M3** and **R-M1** were verified by CD spectroscopy (JASCO Corporation, JC-1500) at room temperature with a cell path length of 1 cm. **M1**, **M2** and **R-M1** were diluted to 100 μM with deionized water at 25 °C and allowed to stand for 12 h for complete assembly. Detail parameter setting were CD scale:200 mdeg/1.0Dod, measure range: 260~190 nm, scanning speed: 500 nm/min, bandwidth: 1.00 nm, and data pitch: 0.5 nm. All spectra were the average of three measurements.

**Fourier transform infrared spectroscopy (FTIR)**. FTIR was used to analyze the secondary structures and intermolecular interactions of **M1**, **M3** and **R-M1**. **M1**, **M3** and **R-M1** were dispersed in distilled water to 100 μM, and the samples were obtained by standing for 12 h. The liquid sample was freeze-dried where the spectrum range was between 4000 and 450 cm$^{-1}$. All the results were averaged among 16 measurements and ATR was used to collect the spectrum.

**TEM analysis**. TEM was used to characterize the morphology of the assemblies of **R-M1** and **M1** after enzyme cleavage using TECNAI G2 20 s-twin electron microscope. The sample (20 μL, 100 μM, made of deionized water) after standing for 12 h was dropped onto the copper mesh. After allowing static for 10 min, the upper liquid was removed and the copper mesh was dyed with 2% uranyl acetate for 1 min. Finally, the surface of the copper mesh was washed with distilled water. The copper mesh was placed overnight at room temperature and then observed.

**Nanofiber dynamic growth**. ThT solution (1.75 mM) was used to detect the amyloid fibers formed by **R-M1** with emission spectrum shift from 440 nm to 490 nm.

**Cytotoxicity assay**. The cytotoxicity of **M1** and **M2** was evaluated using CCK-8 assay in MIA PaCa-2 cells. Cells were seeded into 96-well plates at a density of $5 \times 10^4$ cells per well and placed in an incubator for 14 h. Subsequently, the sample was diluted into a series of different concentrations (0, 50, 100, 150, 200, 250, 300, and 350 μM) and incubated with cells for 24 h. After that, 10 μL of CCK-8 solution was added to each well and incubated for 2 h. The absorbance of blank wells (Ab), sample wells (As) and control wells (Ac) were measured with a microplate reader at a test wavelength of 450 nm and a reference wavelength of 690 nm. The cell viability rate (%) is equal to $(A_s - A_b) / (A_c - A_b) \times 100\%$. All experiments were repeated in triplicate.

**Cellular imaging experiment**. MIA PaCa-2 cells were digested with trypsin, counted with cell counting plate, diluted to $1 \times 10^4$ with DMEM medium, and 1 mL to 1 cm diameter confocal microscope dish was taken. The cells were incubated in 37% $CO_2$ atmosphere for 24 h. The medium containing **M1** and **M2** (50 μM) was replaced and incubated with the cells for 2 h. The cells were washed with PBS for 3 times. CLSM (Zeiss 710) was used to image cells under 40 × objective, excitation = 488 nm, and emission >520 nm.

**Tanswell test**. Transwell migration assay occurred in chemotaxis chambers containing 24 wells. We inoculated cells into the upper chamber in 200 μl DMEM without serum that contained or did not contain ivosidenib. Bottom chambers contained DMEM medium containing 10% FBS. After 24 h of treatment with M1, M2, and M4 (150 μM), we fixed cells using 4% paraformaldehyde and stained with 0.25% crystal violet solution. The stained cells were counted using a microscope.

**In Vivo and ex vivo fluorescence imaging**. All the animal experiments were performed in accordance with the Guide for the Care and Use of Laboratory Animals approved by the Committee for Animal Research of the National Center for Nanoscience and Technology. The MIA PaCa-2 tumor-bearing mice were intravenously injected with **ICG**, **M1** and **M2** (16 mg/kg), respectively for in vivo fluorescence imaging with in vivo spectrum imaging system. Finally, mice were sacrificed to collect the major organs and tumors for ex vivo fluorescence imaging.

**In vivo photoacoustic imaging**. The MIA PaCa-2 tumor-bearing mice were constructed for in vivo photoacoustic imaging of **M1**. After intravenously injected with **M1** (16 mg/kg) through tail vein, the PA images of mice were acquired with MOST (mode: MOST invision 128) at 12 h post injection.

**Establishment of the orthotopic pancreatic cancer xenograft model**. Female BALB/c nude mice (6–8 weeks, 16–18 g) were intraperitoneally anesthetized with Pentobarbital sodium (40 mg/kg body weight) and placed in a lateral position (right side down). After the pancreas was exposed under sterile conditions, MIA PaCa-2 cells resuspended in Matrigel were injected into the parenchyma of pancreas of mice. Subsequently, the body wall and skin incision were successively closed after the pancreas was returned into the peritoneal cavity. After confirming the tumor growth in pancreas, the tumor specific recognition of **M1** and **M2** were investigated with IVIS.

**Tumor slices and staining**. Tumors were harvested and collected in 4% paraformaldehyde solution after tumor-bearing mice treated with **M1** at a dose of 16 mg/kg for 12 h and 48 h. The Hematoxylin and Congo red staining procedure was performed by Google biotechnology (Wuhan) Co. LTD.

**Immunofluorescence and fluorescence imaging**. Orthotopic tumors were dehydration and sliced to 8 μm by Leica 1950 under −22 ℃ after after treated with **M1** at a dose of 16 mg/kg for 48 h. The histologic section was incubated with Anti-CD31 antibody (ab76533, abcam) at 4 ℃ overnight, then washed by TBST and incubated with Goat anti-Rabbit IgG F(ab')2 Secondary antibody, FITC (31573, Thermo Fisher) for 1 h and washed by TBST. The histologic section then treated by antifade mounting medium with Hoechst 33342 (P0133, Beyotime) and imaged by Perkin Elmer Operetta HCS under 20 x and 40 x with 0 or 2% overlaying.

**Toxicology evaluation**. The female Balb/c nude mice (6–8 weeks, $n = 3$) were sacrificed for blood and major organs collection after treated with **M1** at a dose of 16 mg/kg on the 24 h. The histology evaluation of major organs was performed by Google biotechnology (Wuhan) Co. LTD. The blood biochemistry and haematology analyses were carried out by Vital River Laboratory Animal Technology Co. Ltd.

**Statistical methods**. Data are reported as mean ± standard deviation. Statistical analysis of the data was performed with $t$ test followed by post hoc Tukey's test. Statistical significance was defined as $*p < 0.05$, $**p < 0.01$ and $***p < 0.001$.

**Reporting summary**. Further information on research design is available in the Nature Research Reporting Summary linked to this article.

## Data availability

Source data are provided with this paper. All data generated or analyzed during this study are included in this published article and its Supplementary information files. Source data are provided with this paper.

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

## Acknowledgements

This work was supported by the National Key R&D Program of China (2018YFE0205400), National Natural Science Foundation of China (51873045), CAS Interdisciplinary Innovation Team and Youth Innovation Promotion (2017053).

## Author contributions

L.-L. Li, H.R., and X.-X.Z. conceived and designed the project. H.R., X.-X.Z., and X.-Z.Z. carried out the synthesis and characterization of the BIVA probes. H.Y. and L.Z. contributed the molecular dynamics (MD) simulation calculations parts. H.R., X.-Z.Z., and D.-Y.H. performed the fluorescence imaging studies both in vitro and in vivo. L.-L.L., H.-W.X., and H.W. discussed the experimental design. M.Y. helped the revision of language. L.-L.L., H. R., X.-Z.Z., and D.-Y.H. co-wrote the manuscript. All authors interpreted data, provided critical insights, and edited the manuscript.

## Competing interests

The authors declare no competing interests.
