## [Peer Review File · Nature Communications]

REVIEWER COMMENTS

Reviewer #1 (Remarks to the Author): Expert in self assembled probes

The manuscript by Li et al. reported a bioactivated in vivo assembly (BIVA) nanotechnology fabricated NIR probe (M1) for pancreatic tumor imaging. It consists of i) a PEG motif to increase circulation time; ii) a tailoring motif responsive to FAP- α ; iii) a self-assembling motif; iv) IR783 containing imaging motif; and v) an RGD targeting motif. M1 significantly increases the circulation time, actively targets the tumor membrane, and selectively cleaved by FAP- α to induce in situ self-assembly. Replacing the enzyme-responsive motif or substituting the RGD targeting motif undermines the efficacy of M1. M1 selectively targets pancreatic tumors with long retention time, while exhibiting no significant acute toxicity, as indicated by in vivo experiments. I support the acceptance of this manuscript after the authors address the following issues.

1. Please double check the sequence of M2 in line 91. It should be mPEG-AGGKLVFFGC(IR783)GRGD.
2. The FAP- α concentration is 50 mM, which is quite high. Could the authors provide rationale for selecting this concentration for in vitro experiment?
3. It would be beneficial if the authors provide TEM images of M1 without FAP- α cleavage.
4. Will the R-M1-FITC have similar results as M1-FITC when incubated with Miapaca-2 cells?
5. A similar molecule was reported by the same group on *Angewandte Chemie*, 131(43), 15431-15438. The function and application of the reported molecule are like those in this manuscript. So, I suggest the authors to discuss how this work is fundamentally different from the previous report.

Reviewer #2 (Remarks to the Author): Expert in cancer imaging

KEY RESULTS AND SIGNIFICANCE

The manuscript reports 1) the fabrication of a NIR probe for small pancreatic tumor intraoperative navigation imaging by employing a bioactivated in vivo assembly (BIVA) nanotechnology; 2) the structural and conformational characterizations of the BIVA probe, including the MD simulations, CD, FT-IR, fluorescence, and WAXS spectroscopic studies and TEM imaging; 3) the characterizations of the utilities of the probe in the in vitro and in vivo mouse model. The characterizations are comprehensive, significant, and support the main findings that the BIVA probe exhibits a synergetic effect of active targeting and assembly induced retention, resulting in an increased accumulation of the probe at tumor boundary and extended circulation time. In addition, BIVA does not appear to have significant acute toxicity. The reported research is significant for contrast agent/IR fluorescence probe design for pancreatic tumor imaging and has clinical relevance.

VALIDITY

While the reported data support the main findings, some of the interpretations may not be accurate or valid. They need to be addressed.

1) Regarding to Fig. 3a, the manuscript states “implied a β -sheet and α -helix hybrid structure”. However, the CD spectrum for M3 appears to be a α -helical protein structure (Greenfield, N.J. Using circular dichroism spectra to estimate protein secondary structure. Nature protocols 2006, 1, 2876-2890). The spectrum for M1 was interpreted as “a random coil secondary structure in CD spectrum as monomers”. However, a random coil structure should have a negative 195 nm (and a weak positive 210 nm peaks) (Greenfield, N.J., Nature protocols 2006), whereas M1 showed a positive \sim 197 nm peak.

2) The authors claimed 1694 cm^{-1} , 1662 cm^{-1} are two peaks in Fig. 3b. However, these blips/small bumps don't look like a peak in the spectrum. In contrast, in citation 23, 1694 cm^{-1} is clearly a visual peak. Similarly, in Fig. 3d, 1696 cm^{-1} , 1673 cm^{-1} are not apparent peaks, either. Since these samples were freeze-dried, it's unclear why only blips are observed if they're truly signatures. How does a typical single measurement look like?

3) The authors claimed that “Although the targeting group of M2 and BIVA group of M1 had significant interaction with cells, there was no obvious cytotoxicity at high concentration of 300 μM (Supplementary Fig. S17).” However, Fig. S17 only shows that there is no significant difference in cell viability between M1 and M2 incubated cells, and there is no comparison between PBS-treated (control) and M1 or M2-treated?

DATA AND METHODOLOGY

The approach is valid and the data are convincing. Data have been repeated ($n=3$), except for Fig. 5e—unclear how many times it has been repeated and what p value is if it's been repeated multiple times. Most charts/figures are of high quality, i.e. nice presentation and easy to understand. However, there are multiple issues in the labels.

1) In Fig. 3g, the blow-up sub-figure, the label in the figure should be $5.0 \pm 0.4 \text{ nm}$, not $5.0 \pm 0.4 \mu\text{m}$.

2) In Fig. 3e, the x axis label should be min, not “h”.

3) Fig. 4b, labels are confusing, two identical R-M3 in black color but with different retention time.

4) For Fig. 8a&b, $n=?$

5) Fig. S9 & S10 missing y axis labels.

6) Fig. S15 does not have any label.

In addition, some figure legends need improving. For example, Fig. 4 legend should state how many times the experiments were performed ($n=?$); the legend for Fig. 4c should explain what the red arrows are. In Fig. 4e, Tris buffer concentration should be in “mM”, not “mm”; NaCl concentration should be in “M” not “m”.

In the main text, it is stated that CAC was determined based on the Thioflavin T (ThT) analysis but in the supplementary materials, it is stated that a pyrene probe was used to measure CAC. This needs clarification.

SUGGESTED IMPROVEMENTS

The reported data are comprehensive and support the main findings.

CLARITY AND CONTEXT

The overall flow is acceptable. However, a significant portion of the text (or sentences) lacks clarity and is confusing. Please see the annotated manuscript for details.

The introduction section needs to provide sufficient background information (such as the previous works in the relevant fields and how they are specifically related to this research), and clearly/specifically points out the knowledge gap and how this work fills the gap. I have found a nice review and thought it would be helpful for readers: Hu, Z.; Chen, W.-H.; Tian, J.; Cheng, Z. NIRF Nanoprobes for Cancer Molecular Imaging: Approaching Clinic. Trends in Molecular Medicine 2020, 26, 469-482.

In addition, it should also address specifically how this research differs from that reported in citation 20 (Zhao, X.-X., Li, L.-L., Zhao, Y., An, H.-W., Cai, Q., Lang, J.-Y., Han, X.-X., Peng, B., Fei, Y., Liu, H., Qin, H., Nie, G., Wang, H., In Situ Self-Assembled Nanofibers Precisely Target Cancer-Associated Fibroblasts for Improved Tumor Imaging. *Angewandte Chemie International Edition* 58, 15287-15294(2019)), such as what the improvements are.

There were multiple apparent and significant mistakes/typos. For example, in the first paragraph in the Results section, M2 is (mPEG-GPAKLVFFGC(IR783)GRGD), which is exactly the same as M1. Please see the annotated manuscript for more details.

Reviewer #3 (Remarks to the Author): Targeted delivery

The authors developed a fluorescent probe that self assembles in the tumor after it is cleaved by fibroblast activation protein- α (FAP- α) expressed on the membrane of cancer-associated fibroblasts (CAFs). The probe has an innovative design and is well characterized chemically and structurally. However, its biological functions are not properly characterized diminishing the significance of the study.

1. The authors have incorporated a cleavage site in the probe, which reacts to FAP- α expressed on CAFs. While this is a reasonable approach given that pancreatic ductal adenocarcinoma (PDAC) is usually desmoplastic, the authors have not characterized the probe using CAFs or desmoplastic PDAC to assess whether their approach was valid. For example, in Fig. 5c, the authors have used MIA-PaCa-2 PDAC cells instead of CAFs. Do MIA-PaCa-2 cells express FAP- α ? Isn't it more logical to use FAP- α -positive CAFs? There are a few more related concerns. First, why are the MIA PaCa-2 cells round and floating in Fig 5c and 5d? MIA PaCa-2 cells usually attach and spread well in 2D culture. They do not appear healthy as 2D cultures or as 3D spheroids. Second, was the activation indeed caused by FAP- α in the cell culture experiments and in vivo (which relates to the earlier point above)? The authors should establish direct involvement of FAP- α in the activity by using specific inhibitors of FAP- α (or gene editing techniques). In addition, MIA PaCa-2 tumors are usually not that desmoplastic. How much desmoplasia do their tumors have? Do the tumors have CAFs and do they express FAP- α ?

2. It is unclear why it is “expected” that the probe inhibits tumor cell migration (Fig. 5f-g). These data suddenly appear as part of the biological characterization of the probe. The authors neither note any rationale for this experiment nor the methods they have used. In addition, the figure legend for this study (and for many other figures) does not adequately explain the figures making it very difficult to understand the message (see legend for Fig. 5h for example. What is red, what is blue?).

3. It is highly questionable whether the signals circled in Fig. 6a are truly from the orthotopic PDAC. First, the mice are imaged from the right side while orthotopic PDAC resides in the left flank. (In fact, the images provided in Fig. 7a and 7d were taken from the left side of the mice.) Second, the circled signals are deep inside the pelvis where the rectum and urinary bladder are located. The reviewer has a significant concern that the authors are measuring the signals from an irrelevant organ. Under this situation, the data shown in Fig. 6c are also highly skeptical.

4. The reviewer wonders why in vivo tumor imaging was not performed for the mice shown in Fig. 7a and 7d. Only ex vivo imaging data are provided.

5. The authors have used congo red to stain the probes (?) in Fig. 7e and 7f. Congo red is usually used to stain amyloids and certain bacterial elements. The authors have not provided any evidence that congo red stains the probes. Why not directly image the probes on the sections based on the fluorescence? There is also a logical flaw even if the congo red stain does show the location of the probes. Based on the fact that some of the arrows in Fig. 7f appear to point to congo red-positive blood vessels in the tumor (which is not specified either), it appears that the probes do not spread into the extravascular tumor tissue where CAFs are usually found. Then, how was the probe proteolytically activated? Do tumor blood vessels also express FAP- α ?

6. The authors have characterized the FAP- α -dependent cleavage of the probe and the resulting assembly of the probe reasonably well. However, they have not characterized the effect of the RGD motif on tumor-specific targeting at all. They have made M4, a variant of M1 that has a disrupted RGD motif as described under Table 1. However, they have not used M4 in any of the in vitro or in vivo studies to assess the role of RGD in PDAC targeting. The authors need to make sure that the RGD motif is used to target α_v integrins expressed on cultured cells and angiogenic blood vessels at least using M4 and blocking antibodies. Another option is to study RGD-dependent binding of the probe to purified α_v integrins.

7. The authors suggest to use these probes to locate PDAC intraoperatively. However, in real life, it is unlikely that accurate intraop localization of small PDAC makes a whipple procedure or other forms of PDAC resection more precise unless the resection is performed for premalignant lesions that may contain a cryptic malignancy or pancreatic neuroendocrine tumors that are sometimes difficult to identify. Other indications may exist. However, the reviewer feels that the authors have not put in enough thoughts into these points or involved a cancer specialist in their team.

8. Statistical analysis was performed only for Fig. 7b within the main figures. In addition, ANOVA was used while a t-test would be sufficient in this case. No other statistical analyses were performed within the main figures.

9. The reviewer feels that many of the above points have been missed in this study likely because the authors did not verify their data critically enough. It is reflected in the Discussion, which consists of only one paragraph that simply summarizes the data with no insight or discussion whatsoever.

10. It is recommended to have heat-inactivated FAP- α as a control for Fig. 4b to further confirm FAP- α -specific effects.

11. It is highly recommended that the authors utilize an English editing service when rewriting the manuscript. The current manuscript is very difficult to read or understand.

Reviewer #4 (Remarks to the Author): Expert in molecular dynamics

Concerning the Molecular Dynamics (MD) simulations briefly reported in this manuscript, there are a few minor issues that require clarifications concerning both the reported results and their broader significance. Here below I list my comments and suggestions.

1. In the methodological section, concerning the MD simulations, we read that "The topology and force field parameters files for the PEG were generated with Acyppe Antechamber": what about the rest of the molecule? And what is the molecular weight of the simulated PEG moiety? Does it match the experimental value? In the text PEG2000 is mentioned, but judging from Fig. 2 a much shorter chain was actually modelled. Which starting geometries were assumed? Why a temperature of 300 K was selected and not a physiological one (310 K, say)? Finally, at line 437 a timestep of 2 ps is mentioned: most likely, it is a misprint for 2 fs.

2. At line 116, the citation to ref 22 seems to be inappropriate, because the cited paper is a review dealing with MD simulations of proteins at large pressures, so it is not particularly significant for the present study.

3. At lines 120-121 CASP12 is mentioned: why this notation? It should simply be aminoacid ASP12.

4. What is the meaning of the coloured envelopes in the two left-most structures of Fig. 2? And anyway, are these the optimized geometry, or two snapshots taken during the MD run? More importantly, the authors should provide some information about the molecular mobility (or rigidity) at room temperature or at physiological temperature to have a clue about the relevance of the reported β -hairpin conformations.

5. At line 148, on the basis of circular dichroism spectra the authors find "well-ordered β -sheet assembled secondary structures", Indeed, these are somehow consistent with the MD simulations: in fact, at line 130 the author anticipate that the conformation of R-M1 would be "beneficial to the

occurrence of intermolecular dynamic assembly". Simulation of this intermolecular assembly were not performed, but it is quite possible that the assembly (or aggregation) may enhance the β -sheet ordering: could the authors briefly comment on this issue?

Title: "A bioactivated in vivo assembly (BIVA) nanotechnology fabricated NIR probe for small pancreatic tumor intraoperative navigation imaging" (Manuscript ID: NCOMMS-21-15472)

Our point-by-point responses to reviewers:

Reviewer: 1

Comments: The manuscript by Li et al. reported a bioactivated in vivo assembly (BIVA) nanotechnology fabricated NIR probe (M1) for pancreatic tumor imaging. It consists of i) a PEG motif to increase circulation time; ii) a tailoring motif responsive to FAP- α ; iii) a self-assembling motif; iv) IR783 containing imaging motif; and v) an RGD targeting motif. M1 significantly increases the circulation time, actively targets the tumor membrane, and selectively cleaved by FAP- α to induce in situ self-assembly. Replacing the enzyme-responsive motif or substituting the targeting motif undermines the efficacy of M1. M1 selectively targets pancreatic tumors with long retention time, while exhibiting no significant acute toxicity, as indicated by in vivo experiments. **I support the acceptance of this manuscript after the authors address the following issues.**

1. Please double check the sequence of M2 in line 91. It should be mPEG-AGGKLVFFGC(IR783)GRGD.

Response: Thanks for reminding. We are sorry for the writing error about the sequence of M2, the right sequence is mPEG-AGGKLVFFGC(IR783)GRGD. We have double checked the whole manuscript and highlighted the revision.

2. The FAP- α concentration is 50 mM, which is quite high. Could the authors provide rationale for selecting this concentration for in vitro experiment?

Response: Thanks for the comments. It's against common sense and quite high with FAP- α in cleavage experiments. We found the experiment steps in handwriting record is 'FAP- α buffer: 1 mL 50 mM Tris, 1M NaCl, 1 mgmL⁻¹ BSA, pH 7.5; final concentration of M1 or M4: 100 μ M; Cultured conditions: with 1 μ L 50 mM FAP- α overnight', which means the concentration of FAP- α is 50 μ M. Sorry again for the clerical error, and we had corrected the related paragraphs in revised manuscript.

3. It would be beneficial if the authors provide TEM images of M1 without FAP- α cleavage.

Response: Thanks for the comments. We had supplemented a TEM experiment about M1 without FAP- α cleavage. Experiment steps were as the same as M1 cleavage, the only different is that FAP- α was replaced by an inactivated one. As shown in Supplementary Fig. 16, the spots in the image are FAP- α , we didn't observe any fiber or ribbon structure and the solution of M1 with inactivated FAP- α was clear. The experiments confirmed that M1 cannot assembly into nanofiber even with inactivated FAP- α .

Supplementary Fig. 16 | TEM image of M1 co-incubated with inactivated FAP- α for 12h.

4. Will the R-M1-FITC have similar results as M1-FITC when incubated with Miapaca-2 cells?

Response: Thanks for question. We had supplemented the incubation of R-M1-FITC with Miapaca-2 cells under the same condition. As showed in Supplementary Fig. 19, the R-M1-FITC as the enzyme tailored residues was assembled originally before administration to cells. Thus, the results showed that the assembled R-M1-FITC showed weak signal on the cell membrane with few nonspecific adherence. While, the M1-FITC was active targeting and in situ FAP- α tailored to induce assembly, which exhibited typical colocalization on the membrane (Fig. 5d). The two molecules showed significant difference on the cell membrane bonding capability, which mean that the pre-assembly and in situ triggered assembly dramatically affect the cell membrane interactions.

Supplementary Fig. 19 | CLSM images of R-M1 incubated Miapaca-2 cells after incubation for 1 h without washing.

5. A similar molecule was reported by the same group on *Angewandte Chemie*, 131(43), 15431-15438. The function and application of the reported molecule are like those in this

manuscript. So, I suggest the authors to discuss how this work is fundamentally different from the previous report.

Response: Thanks for the advice. Our group designed a peptide based fluorescent probe in 2019 (Angew. Chem. Inter. Ed. 131, 15431). In this work, we have a great improvement based on our previous work. There are four detailed explanations for the improvement as below. The discussion of the difference and improvement are added in the revised manuscript.

1. In our previous work, we modular designed an enzyme-responsive self-assembled peptide probe to realize the AIR (assembly induced retention) effect from molecules to assemblies, resulting an enhanced retention for long-term imaging. On this basis, we proposed a new strategy named as bioactivated in vivo assembly (BIVA), which included a new targeting mechanism based on dynamic synergy between active targeting and AIR effect. Compare with previous work, the BIVA technology improved the tumor imaging specificity and sensitivity.

2. In our new work, the molecule design added a new PEG tail. There are two significant reasons. Firstly, we found that the PEG tail affected the hydrophilic-hydrophobic balance and the molecule conformation, which was easy for design the BIVA system. According to our MD simulation, we clearly understand the conformation transformation before/after enzyme cleavage. Secondly, the PEG tail enables BIVA probes to achieve long-time blood circulation for a better further pharmacokinetics behavior.

3. This manuscript firstly linked BIVA probe with in vivo pharmacokinetics, and further verifies the feasibility of BIVA probe at the in vivo level through the analysis of pharmacokinetics.

4. Unlike before, this article chooses the intraoperative navigation for pancreatic cancer as the application. Early pancreatic cancer tumors are small and difficult to image. Additionally, pancreatic is a deep organ, and Fap- α overexpression in pancreatic cancer cells provide a more challenging research model.

Reviewer: 2

Comments: The manuscript reports 1) the fabrication of a NIR probe for small pancreatic tumor intraoperative navigation imaging by employing a bioactivated in vivo assembly (BIVA) nanotechnology; 2) the structural and conformational characterizations of the BIVA probe, including the MD simulations, CD, FT-IR, fluorescence, and WAXS spectroscopic studies and TEM imaging; 3) the characterizations of the utilities of the probe in the in vitro and in vivo mouse model. The characterizations are comprehensive, significant, and support the main findings that the BIVA probe exhibits a synergetic effect of active targeting and assembly induced retention, resulting in an increased accumulation of the probe at tumor boundary

and extended circulation time. In addition, BIVA does not appear to have significant acute toxicity. **The reported research is significant for contrast agent/IR fluorescence probe design for pancreatic tumor imaging and has clinical relevance.**

1. Regarding to Fig. 3a, the manuscript states “implied a β -sheet and α -helix hybrid structure”. However, the CD spectrum for M3 appears to be a α -helical protein structure (Greenfield, N.J. Using circular dichroism spectra to estimate protein secondary structure. Nature protocols 2006, 1, 2876-2890). The spectrum for M1 was interpreted as “a random coil secondary structure in CD spectrum as monomers”. However, a random coil structure should have a negative 195 nm (and a weak positive 210 nm peaks) (Greenfield, N.J., Nature protocols 2006), whereas M1 showed a positive \sim 197 nm peak.

Response: Thank for the advice of reviewer. We have repeated the CD spectra of M1 and M3 in Fig. 3a. The distribution of the secondary structures of M1, M3 and R-M1 were also calculated according to the Reed’s Reference (Fig 3e and Supplementary Table 1). The results indicated that the secondary structure of M1 is mainly consist of Random Coil (66.1 %), but M3 are consist with Helix, Beta, and Random Coil (45 %, 38.7 %, and 16.3%), as R-M1 are mainly consisted by Beta Sheet (72 %). The results infer us that the conjugation of IR783 could significantly influence the conformation of M1 and M3. The M3 molecules without IR783 labeling exhibited a β -sheet assembled behavior, while the mainly random coil conformation of M1 was still as monomers. While, remove the PEG tail from M1, the residues of R-M1 could self-assembly from oligomer to highly ordered anti-parallel β -sheet structures. All the new data were added in the revised manuscript in Fig. 3a, Fig. 3d, Fig. 3e, and Supplementary Table 1 as well as below.

Fig. 3 | Assembled structure conformations and self-assembly behavior in aqueous solution. a, CD spectra of M1 and M3 in DI water under a concentration of 200 μ M. b, The FTIR spectra of M1 and M3. Anti-parallel β -sheet characteristic peaks (green arrows); Parallel β -sheet characteristic peaks (blue arrows); α -Helix structure (yellow arrow). c, The typical β -sheet CD spectra of R-M1 in DI water (insert figure: the Dunder phenomenon) under a concentration of 100 μ M. d, The FTIR spectra of dynamic growth of R-M1. e, Analysis of secondary structure composition of M1, M3, and R-M1 based on CD spectra. f, Elongation-nucleation growth procedure with ThT staining. g, The WAXS spectrum and illustration of the R-M1 fibrils. h, The TEM images of nanofibers morphology of R-M1. Data: mean \pm standard deviation (n=3).

Supplementary Table 1 | Secondary structure proportions of M1, M3 and R-M1 calculated by Reed's Reference based on CD spectra.

	M1	M3	R-M1
Helix	0	45	0.5
Beta	32.2	38.7	72
Turn	1.7	0	27.5
Random	66.1	16.3	0
RMS	14.093	5.66	23.908

2. The authors claimed 1694 cm^{-1} , 1662 cm^{-1} are two peaks in Fig. 3b. However, these blips/small bumps don't look like a peak in the spectrum. In contrast, in citation 23, 1694 cm^{-1} is clearly a visual peak. Similarly, in Fig. 3d, 1696 cm^{-1} , 1673 cm^{-1} are not apparent peaks, either. Since these samples were freeze-dried, it's unclear why only blips are observed if they're truly signatures. How does a typical single measurement look like?

Response: Thanks for reminding. We had repeat the FTIR results for M3 and R-M1, results were showed in Fig. 3b and Fig. 3d. The 1647 cm^{-1} of the random structure derived by M1 in Fig. 3b can be clearly observed. The peaks at 1629 cm^{-1} , 1675 cm^{-1} , and 1698 cm^{-1} of M3 indicated anti-parallel β -sheet structure (green arrows), peaks at 1648 cm^{-1} and 1663 cm^{-1} indicated parallel beta sheet structure (blue arrows), the exist of 1654 cm^{-1} (yellow arrow) indicated α -Helix structure in M1 which verified the results deduced by CD in Fig 3a. As refer

in manuscript, the peak at 1634 cm^{-1} of R-M1 of 1 min sample were indicated as oligomer, the 1698 cm^{-1} , 1688 cm^{-1} , and 1629 cm^{-1} were all assigned to anti-parallel β -sheet. (J. Am. Chem. Soc, 1961, 83, 712. Chem. Commun., 2014, 8923.) All the FTIR results firmly confirm the secondary structures content indicated by CD. The new data were added in Fig. 3b and Fig. 3d in the revised manuscript as below.

Fig. 3 | Assembled structure conformations and self-assembly behavior in aqueous solution. **a**, CD spectra of M1 and M3 in DI water under a concentration of $200\ \mu\text{M}$. **b**, The FTIR spectra of M1 and M3. Anti-parallel β -sheet characteristic peaks (green arrows); Parallel β -sheet characteristic peaks (blue arrows); α -Helix structure (yellow arrow). **c**, The typical β -sheet CD spectra of R-M1 in DI water (insert figure: the Durdal phenomenon) under a concentration of $100\ \mu\text{M}$. **d**, The FTIR spectra of dynamic growth of R-M1. **e**, Analysis of secondary structure composition of M1, M3, and R-M1 based on CD spectra. **f**, Elongation-nucleation growth procedure with ThT staining. **g**, The WAXS spectrum and illustration of the R-M1 fibrils. **h**, The TEM images of nanofibers morphology of R-M1. Data: mean \pm standard deviation ($n=3$).^{††}

3. The authors claimed that “Although the targeting group of M2 and BIVA group of M1 had significant interaction with cells, there was no obvious cytotoxicity at high concentration of $300\ \mu\text{M}$ (Supplementary Fig. S17).” However, Fig. S17 only shows that there is no significant difference in cell viability between M1 and M2 incubated cells, and there is no comparison between PBS-treated (control) and M1 or M2-treated?

Response: Thank you for your suggestions. We re-verified the cytotoxicity test of M1 and M2 and added the control group. The experimental results are shown in Supplementary Fig. 24. According to this figure, we found that M1 and M2 have no obvious cytotoxicity below $300\ \mu\text{M}$. The results of this experiment are like our previous results and are reproducible.

Supplementary Fig. 24 | Cell viability assay of Miapaca-2 cells treated with a series concentration of **M1** and **M2** for 24 h. Data were expressed as mean \pm SD (n=6).

DATA AND METHODOLOGY

The approach is valid and the data are convincing. Data have been repeated (n=3), except for Fig. 5e—unclear how many times it has been repeated and what p value is if it's been repeated multiple times. Most charts/figures are of high quality, i.e. nice presentation and easy to understand. However, there are multiple issues in the labels.

1. In Fig. 3g, the blow-up sub-figure, the label in the figure should be 5.0 ± 0.4 nm, not 5.0 ± 0.4 μ m.

Response: Thanks for reminding. We had corrected it to 5.0 ± 0.4 nm.

2. In Fig. 3e, the x axis label should be min, not "h".

Response: Thanks for reminding. We had changed the "h" to "min" in Fig. 3e.

3. Fig. 4b, labels are confusing, two identical R-M3 in black color but with different retention time.

Response: Thanks for advice. We had changed the color from black to red and green for better distinction to readers.

4. For Fig. 8a&b, n=?

Response: We are sorry and we supplement the n=4.

5. Fig. S9 & S10 missing y axis labels.

Response: Thanks for reminding, and we had added the y axis labels below.

6. Fig. S15 does not have any label.

Response: Thanks for kind reminding. We had added the label and bars in Fig. S15.

In addition, some figure legends need improving. For example, Fig. 4 legend should state how many times the experiments were performed (n=?); the legend for Fig. 4c should explain what the red arrows are. In Fig. 4e, Tris buffer concentration should be in “mM”, not “mm”; NaCl concentration should be in “M” not “m”.

In the main text, it is stated that CAC was determined based on the Thioflavin T (ThT) analysis but in the supplementary materials, it is stated that a pyrene probe was used to measure CAC. This needs clarification.

Response: Thanks for reminding, sorry for the mistake again. We had corrected the experiment steps about CAC measurement in manuscript after checked the original experiment record book.

SUGGESTED IMPROVEMENTS

The reported data are comprehensive and support the main findings.

CLARITY AND CONTEXT

The overall flow is acceptable. However, a significant portion of the text (or sentences) lacks clarity and is confusing. Please see the annotated manuscript for details.

The introduction section needs to provide sufficient background information (such as the previous works in the relevant fields and how they are specifically related to this research), and clearly/specifically points out the knowledge gap and how this work fills the gap. I have found a nice review and thought it would be helpful for readers: Hu, Z.; Chen, W.-H.; Tian, J.; Cheng, Z. NIRF Nanoprobes for Cancer Molecular Imaging: Approaching Clinic. Trends in Molecular Medicine 2020, 26, 469-482.

In addition, it should also address specifically how this research differs from that reported in citation 20 (Zhao, X.-X.,Li, L.-L.,Zhao, Y.,An, H.-W.,Cai, Q.,Lang, J.-Y.,Han, X.-X.,Peng, B.,Fei, Y.,Liu, H.,Qin, H.,Nie, G.,Wang, H., In Situ Self-Assembled Nanofibers Precisely Target Cancer-Associated Fibroblasts for Improved Tumor Imaging. Ange Chem Inter Ed. 58, 15287-15294(2019)), such as what the improvements are.

There were multiple apparent and significant mistakes/typos. For example, in the first paragraph in the Results section, M2 is (mPEG-GPAKLVFFGC(IR783)GRGD), which is exactly the same as M1. Please see the annotated manuscript for more details.

Response: Thank you for your suggestions. We have checked and modified the writing and spelling as much as possible, changed some descriptions the introduction and main body of the article. All the modified parts are highlighted.

Our group designed a peptide based fluorescent probe in 2019 (Angew. Chem. Inter. Ed. 131, 15431). In this work, we have a great improvement based on our previous work. There are four detailed explanations for the improvement as below.

1. In our previous work, we modular designed an enzyme-responsive self-assembled peptide probe to realize the AIR (assembly induced retention) effect from molecules to assemblies, resulting an enhanced retention for long-term imaging. On this basis, we proposed a new strategy named as bioactivated in vivo assembly (BIVA), which included a new targeting mechanism based on dynamic synergy between active targeting and AIR effect. Compare with previous work, the BIVA technology improved the tumor imaging specificity and sensitivity.

2. In our new work, the molecule design added a new PEG tail. There are two significant reasons. Firstly, we found that the PEG tail affected the hydrophilic-hydrophobic balance and the molecule conformation, which was easy for design the BIVA system. According to our MD simulation, we clearly understand the conformation transformation before/after enzyme cleavage. Secondly, the PEG tail enables BIVA probes to achieve long-time blood circulation for a better further pharmacokinetics behavior.

3. This manuscript firstly linked BIVA probe with in vivo pharmacokinetics, and further verifies the feasibility of BIVA probe at the in vivo level through the analysis of pharmacokinetics.

5. Unlike before, this article chooses the intraoperative navigation for pancreatic cancer as the application. Early pancreatic cancer tumors are small and difficult to image. Additionally, pancreatic is a deep organ, and Fap- α overexpression in pancreatic cancer cells provide a more challenging research model.

Reviewer: 3

Comments: The authors developed a fluorescent probe that self assembles in the tumor after it is cleaved by fibroblast activation protein- α (FAP- α) expressed on the membrane of cancer-associated fibroblasts (CAFs). The probe has an innovative design and is well characterized chemically and structurally. However, its biological functions are not properly characterized diminishing the significance of the study.

1. The authors have incorporated a cleavage site in the probe, which reacts to FAP- α expressed on CAFs. While this is a reasonable approach given that pancreatic ductal adenocarcinoma (PDAC) is usually desmoplastic, the authors have not characterized the probe using CAFs or desmoplastic PDAC to assess whether their approach was valid. For example, in Fig. 5c, the authors have used MIA-PaCa-2 PDAC cells instead of CAFs. Do MIA-PaCa-2 cells express FAP-

α ? Isn't it more logical to use FAP- α -positive CAFs? There are a few more related concerns. First, why are the MIA PaCa-2 cells round and floating in Fig 5c and 5d? MIA PaCa-2 cells usually attach and spread well in 2D culture. They do not appear healthy as 2D cultures or as 3D spheroids. Second, was the activation indeed caused by FAP- α in the cell culture experiments and in vivo (which relates to the earlier point above)? The authors should establish direct involvement of FAP- α in the activity by using specific inhibitors of FAP- α (or gene editing techniques). In addition, MIA PaCa-2 tumors are usually not that desmoplastic. How much desmoplasia do their tumors have? Do the tumors have CAFs and do they express FAP- α ?

Response: Thank reviewer for reminding. We had investigated the expression of FAP- α in pancreatic tumor in Web of Science, PubMed, and Google Scholar in detail.

i). We found that both CAFs cells and MIA PaCa-2 cells in pancreatic tumor overexpress FAP- α (World Journal of Gastroenterol, 2012; 18(8): 840-846).

ii). In previous work, we have improved our strategy on CAFs cells. But we hope to conduct research on pancreatic tumor imaging in this creative work, so we chose MIA PaCa-2 cells for research.

iii). The MIA PaCa-2 appears round or fusiform in different seeding time. We had confirmed that MIA PaCa-2 appears mainly round before culture for 36 hours (Pictures for MIA PaCa-2 in different seeding time were show below to provide a visual evidence for reviewer). We choose to incubate molecules with MIA PaCa-2 after seeding the cell to confocal cell culture dish for 12 h to present a better 3D imaging experiment to show the M1 adhere the whole cell.

iv). We had supplemented the incubation experiment of M1 and MIA PaCa-2 under the RGD and FAP- α inhibitor in Supplementary Fig. 21 and Fig. 22 in the revised SI file. We found that after RGD treatment, the assemblies could observe non-specific interaction on the membrane. While, pre-treated by FAP- α inhibitors, only small amounts of fluorescence could be observed on the cell membrane. This may be because **M1** actively targets cell membranes under the influence of RGD. The cell membrane bonding and retention of the M1 signals were significantly reduced, which validated that the RGD recognition and FAP- α induced nanofibers formation both contributed to the attach on the membranes.

Supplementary Fig. 21 | CLSM images of **M1** incubated in integrins blocked Miapaca-2 cells for 1 h and washed by fresh DMEM for three times (Miapaca-2 cells were treated by RGD for 1h at 50 μ M and washed by fresh DMEM for three times before adding **M1**).

Supplementary Fig. 22 | CLSM images of **M1** incubated in Fap- α inhibited Miapaca-2 cells for 1h and washed by fresh DMEM for three times (Miapaca-2 cells were treated by Fap- α inhibitor Ac-Gly-BoroPro for 1h at 50 nM and washed by fresh DMEM for three times before adding **M1**).

2. It is unclear why it is “expected” that the probe inhibits tumor cell migration (Fig. 5f-g). These data suddenly appear as part of the biological characterization of the probe. The authors neither note any rationale for this experiment nor the methods they have used. In addition, the figure legend for this study (and for many other figures) does not adequately explain the figures making it very difficult to understand the message (see legend for Fig. 5h for example. What is red, what is blue?).

Response: Thanks for suggestion. This experiment aims to explore the effect of M1 molecules on tumor cells, which try to clarify the role of shearing and targeting modules. According to the experimental results, we speculate M1 can be more enriched in the tumor site and form fibers through assembly to inhibit tumor cell migration compare to M2 and M4. While, M2 and M4 have no ability to target and adhere to cell membrane. There are cases where similar assemblies affect tumor cell migration (CHINESE CHEMICAL LETTERS, 2020; 31 (7): 1787). According to the experimental results, we found that the M1 molecule has a strongest inhibitory through the comparison of M2 and M4 molecules effect on the migration of tumor cells.

Sorry for confusing, we added the caption in Fig. 5h. The visual pie chart shows the results of the tumor migration experiment. The blue part is the transferred cells, compared with the

transferred cells of Blank (100%). We also added the experiment methods in the revised manuscript as below.

Fig. 5 | The BIVA effect with enhanced targeting to pancreatic tumor cell for outline location imaging. a, Illustration of BIVA effect: active targeting and assembly/aggregate induced retention (AIR) effect. b, Chemical structure of FITC labeling. c, 2D and 3D confocal images of M2 with active targeting property and M1 with BIVA effect on MIA PaCa-2 cells after incubation for 1 h. d, The distribution of M1 on MIA PaCa-2 cell. e, ThT staining of the lysed cell membrane of MIA PaCa-2. f, Illustration of migration inhibition after treated by BIVA probe. g, The image of the transferred cells after treatment of PBS (blank), M2, M4, and M1. h, The pie diagram of quantitative statistical calculation of the transferred cells (blue) in different groups. The blank control was set as 100% transferred cells.

Transwell test: Transwell migration assay occurred in chemotaxis chambers containing 24 wells. We inoculated cells into the upper chamber in 200 μ l DMEM without serum that contained or did not contain ivosidenib. Bottom chambers contained DMEM medium containing 10% FBS. After 24 h of treatment with M1, M2, and M4 (150 μ M), we fixed cells using 4% paraformaldehyde and stained with 0.25% crystal violet solution. The stained cells were counted using a microscope.

3. It is highly questionable whether the signals circled in Fig. 6a are truly from the orthotopic PDAC. First, the mice are imaged from the right side while orthotopic PDAC resides in the left flank. (In fact, the images provided in Fig. 7a and 7d were taken from the left side of the mice.) Second, the circled signals are deep inside the pelvis where the rectum and urinary bladder are located. The reviewer has a significant concern that the authors are measuring the signals from an irrelevant organ. Under this situation, the data shown in Fig. 6c are also highly skeptical.

Response: Sorry for the confusing. It should be clarified that the circled tumor position in Fig. 6a is the **subcutaneous tumor** of the right hind leg of the mouse, and the sideways position is used for shooting. There are two main reasons why we used the subcutaneous tumor model in Fig. 6 to observe the imaging characteristics and perform pharmacokinetic statistics of each molecule. Firstly, the subcutaneous tumor model of the right hind leg does not overlap with other organs, and it can reduce the organs depth difference caused system error. Secondly, compared with the orthotopic tumor, the size of the subcutaneous tumor is more controllable, reducing the individual difference between the experimental groups, and making the experimental data more reliable. After clearly understand our BIVA probe's distribution and

elimination behavior. We validated our BIVA probe on orthotopic PDAC mouse model (Fig. 7) to verify the highly specific small-sized tumor imaging and enhanced imaging contrast for intraoperative navigation. We have clearly described the tumor model in our previous manuscript (seen the revision as below), just to make sure there's no ambiguity.

Metabolic difference and optimized biodistribution enhanced imaging. In order to reduce systemic error and individual difference, we constructed the subcutaneous pancreatic cancer model in mice for quantitatively calculating of the metabolic data. Firstly, the subcutaneous tumor model of the right hind leg had no overlap with other organs, and it can reduce the organs depth difference caused system error. Secondly, compared with the orthotopic tumor, the size of the subcutaneous tumor was more controllable, reducing the individual difference between the experimental groups, and making the experimental data more reliable. According to the time-dependent *in vivo* NIR images, there were significant differences in the fluorescence distribution

4. The reviewer wonders why *in vivo* tumor imaging was not performed for the mice shown in Fig. 7a and 7d. Only *ex vivo* imaging data are provided.

Response: We are sorry for the data display. We have performed *in vivo* imaging experiments, but we didn't align it for presentation, because we think this part of the data is not so important. According to the reviewer's suggestion, we provide the *in vivo* tumor imaging for the mice in Fig. 7a as supplementary materials in the added Supplementary Fig. 26. Furthermore, we also provided the *in vivo* distribution of M1 and M2 after 48 h administration in Supplementary Fig. 27. All the new data can be found in the highlighted manuscript and SI.

Supplementary Fig. 26 | Images of M1 and M2 molecules *in situ* in nude mice with pancreatic cancer were measured at 12 h post injection.

Supplementary Fig. 27 | The *in vivo* and *in situ* NIR images of orthotopic pancreatic tumors by M1 and M2, and the *ex vivo* of organ biodistribution including heart, liver, spleen, lung, kidney, and tumor after vein injection at 16mg/kg for 48 h.

5. The authors have used congo red to stain the probes (?) in Fig. 7e and 7f. Congo red is usually used to stain amyloids and certain bacterial elements. The authors have not provided any evidence that congo red stains the probes. Why not directly image the probes on the sections based on the fluorescence? There is also a logical flaw even if the congo red stain does show the location of the probes. Based on the fact that some of the arrows in Fig. 7f appear to point to congo red-positive blood vessels in the tumor (which is not specified either), it appears that the probes do not spread into the extravascular tumor tissue where CAFs are usually found. Then, how was the probe proteolytically activated? Do tumor blood vessels also express FAP- α ?

Response: Sorry for the confusing. We have provided new data for the tumor slice. We tagged the blood vessels in the vascular tumor area with immunofluorescence using CD31 (green), and the nucleus of tumor cells were stained by hochest3342, and the red color were from M1. As seen in Fig. 7f, most of the M1 were far away from the green colored blood vessels and the signals were uniform distributed in tumor. The whole tumor section both confirmed the conclusion above (Supplementary Fig. S30). As known, Congo red probes were used to locate the amyloids, which could also be used for staining our assembled nanofibers. Once stained by Congo Red (Supplementary Fig. S31), the tumor slices were easily observed the red distribution, which mean that the deep penetrated M1 were transformed to nanofibers. All the new revision can be found in highlighted manuscript and SI.

Supplementary Fig. 30 | Fluorescence images of tumor histologic section treated with **M1** (16 mg/kg) for 48 h. Blue: Hoechst 33342, Red: Cy3 labeled **M1**.

Supplementary Fig. 31 | The tumor histologic section in mice treated with **M1** (16 mg/kg) for 48 h with Congo red (red) stained assembled fibrils. A enlarged images were respectively corresponded to the red box.

Fig. 7 | The BIVA probe enhance orthotopic pancreatic tumor imaging *in vivo*. **a**, The *in vivo* NIR images of small orthotopic pancreatic tumor by **M1** and **M2**, and the *ex vivo* of organ biodistribution including heart, liver, spleen, lung, kidney, and tumor. $n=3$ **b**, The quantitative analysis of average fluorescence intensity per organ area. **c**, The *in vivo* NIR images of orthotopic pancreatic tumor with individual difference with the same *i.v.* dose administration for 12 h ($n=5$). **d**, The small size (~ 2 mm diameter) orthotopic pancreatic tumor images and its *ex vivo* signal distribution. **e**, The quantitative calculation of the signal in tumor area and healthy spleen area ($n=8$). **f**, The tumor histochemical staining with Hoechst33342 (blue), CD31 (green) and **M1** (red) post *i.v.* injection of **M1** for 48 h. The yellow arrows pointed at the blood vessels. Bars of the up layer: 200 μm ; Bars of the bottom layer: 50 μm . Data: mean \pm standard deviation. Injection dose (*i.v.* administration): 16 mg/kg. Statistical analysis: one-way **t-test** followed by post hoc Tukey's test, *** $p < 0.001$.

6. The authors have characterized the FAP- α -dependent cleavage of the probe and the resulting assembly of the probe reasonably well. However, they have not characterized the effect of the RGD motif on tumor-specific targeting at all. They have made **M4**, a variant of **M1** that has a disrupted RGD motif as described under Table 1. However, they have not used **M4** in any of the *in vitro* or *in vivo* studies to assess the role of RGD in PDAC targeting. The authors need to make sure that the RGD motif is used to target α_v integrins expressed on cultured cells and angiogenic blood vessels at least using **M4** and blocking antibodies. Another option is to study RGD-dependent binding of the probe to purified α_v integrins.

Response: Thanks for reminding. We had supplemented the incubation of M4-FITC with Miapaca-2 cells under the same condition of M1 and M2. The main different between M4 and M1 is the exist of RGD. As showed in Supplementary Fig. S20, the M4 only showed weak signal on the cell membrane with few nonspecific adherences, which indicated that the RGD motif played an important role for targeting. In contrast, the cell membrane bonding capability of M2 with RGD motif was enhanced to that of M4, which can be seen in Fig. 5c. But, based on BIVA technology in Fig. 5c, the dynamic synergy of active targeting and AIR effect exhibited a significant enhancement of cell membrane bonding and retention.

In order to verify the influence of the addition of the targeting module of the probe molecules, we explored the imaging of R-M1 and M4 on cells. After co-incubation with R-M1, which was an insoluble suspended mixture, the fluorescence signal could be observed both in medium and on the cell membrane (Supplementary Fig. 19). While the cells were treated by M4, only a small amount of fluorescence signal could be observed on cell membrane (Supplementary Fig. 20). The phenomenon observed by R-M1 indicated that with RGD motif, the assembled R-M1 still remain the specific bonding capability to cell membrane. Without RGD bonding, the M4 molecules could also be tailored by FAP- α and the assembled nanofibers exhibited non-specific interaction on the membranes. Moreover, once pre-treated the cells with RGD peptides (Supplementary Fig. 21) or FAP- α inhibitors (Supplementary Fig. 22), the cell membrane bonding and retention of the M1 signals were significantly reduced, which validated that the RGD recognition and FAP- α induced nanofibers formation both contributed to the attach on the membranes.

Supplementary Fig. 19 | CLSM images of R-M1 incubated Miapaca-2 cells after incubation for 1 h without washing.

Supplementary Fig. 20 | CLSM images of M4 incubated Miapaca-2 cells after incubation for 1 h and washed by fresh DMEM for three times.

Supplementary Fig. 21 | CLSM images of M1 incubated in integrins blocked Miaapaca-2 cells for 1 h and washed by fresh DMEM for three times (Miaapaca-2 cells were treated by RGD for 1h at 50 μ M and washed by fresh DMEM for three times before adding M1).

Supplementary Fig. 22 | CLSM images of M1 incubated in Fap- α inhibited Miaapaca-2 cells for 1h and washed by fresh DMEM for three times (Miaapaca-2 cells were treated by Fap- α inhibitor Ac-Gly-BoroPro for 1h at 50 nM and washed by fresh DMEM for three times before adding M1).

7. The authors suggest to use these probes to locate PDAC intraoperatively. However, in real life, it is unlikely that accurate intraop localization of small PDAC makes a whipple procedure or other forms of PDAC resection more precise unless the resection is performed for premalignant lesions that may contain a cryptic malignancy or pancreatic neuroendocrine tumors that are sometimes difficult to identify. Other indications may exist. However, the reviewer feels that the authors have not put in enough thoughts into these points or involved a cancer specialist in their team.

Response: Thanks for referee's comments. In this work, we address a new probe based on bioactivated in vivo assembly (BIVA) technology. The key point here is that our proposed BIVA technology and proved new enhanced targeting mechanism could be used for small-sized orthotopic tumor imaging. We just choose the intraoperative navigation for pancreatic cancer as a demo application. There are three reasons for tumor intraoperative navigation imaging. One is that small sized tumor is difficult observe and easy to ignore during surgery. The other one is that pancreatic is a deep metabolic organ, the specific imaging on such organ is difficult because the low contrast. Finally, FAP- α is overexpression in pancreatic cancer cells, such as MIA PaCa-2, provided us a suitable candidate for research. Of course, such a general technology and targeting mechanism also could be used for other tumors based on changing the specific targeting motif and enzyme tailoring motif. Thank again for this valuable suggestion. We will further deep discuss with cancer specialists for a more suitable application place in our further work.

8. Statistical analysis was performed only for Fig. 7b within the main figures. In addition, ANOVA was used while a t-test would be sufficient in this case. No other statistical analyses were performed within the main figures.

Response: Thanks for advice. We performed T-test analysis on the data involved in Fig. 7b, and the $P = 0.000844 < 0.001$.

9. The reviewer feels that many of the above points have been missed in this study likely because the authors did not verify their data critically enough. It is reflected in the Discussion, which consists of only one paragraph that simply summarizes the data with no insight or discussion whatsoever.

Response: Thank for this suggestion. We have revised and added the detailed discussion to emphasize our deep understanding and conclusion for the data. All the revisions have been highlighted in the revised manuscript.

10. It is recommended to have heat-inactivated FAP- α as a control for Fig. 4b to further confirm FAP- α -specific effects.

Response: Thanks for reminding. We have added the heat-inactivated FAP- α as a control for Fig. 4b in the revised manuscript in Supplementary Fig. 14 and as below. As shown in HPLC curve, after heat-inactivated FAP- α , there has no molecule tailoring of M3, which confirm the FAP- α -specificity.

Supplementary Fig. 14 | HPLC curve of M3 co-incubated with inactivated FAP- α for 12h.

11. It is highly recommended that the authors utilize an English editing service when rewriting the manuscript. The current manuscript is very difficult to read or understand.

Response: Thank reviewer for suggestions. We have tried our best to write articles in rigorous English with high standard. We also ask for a help of an English native speaker. We will consult experts to modify the language of the article as much as possible.

Reviewer: 4

Comments: Concerning the Molecular Dynamics (MD) simulations briefly reported in this manuscript, there are a few minor issues that require clarifications concerning both the reported results and their broader significance. Here below I list my comments and suggestions.

1. In the methodological section, concerning the MD simulations, we read that " The topology and force field parameters files for the PEG were generated with Acyppe Antechamber": what about the rest of the molecule? And what is the molecular weight of the simulated PEG moiety? Does it match the experimental value? In the text PEG2000 is mentioned, but judging from Fig. 2 a much shorter chain was actually modelled. Which starting geometries were assumed? Why a temperature of 300 K was selected and not a physiological one (310 K, say)? Finally, at line 437 a timestep of 2 ps is mentioned: most likely, it is a misprint for 2 fs.

Response: Thanks for referee's important comments. The molecular dynamics simulations of amino acid using the AMBER99SB-ILDN force field. The topology and force field parameters files for the PEG and the rest of the molecule were generated with AmberTools and Acyppe Antechamber. We have modified the corresponding description as "The topology and force field parameters files for the PEG and the rest of the molecule were generated with AmberTools and Acyppe Antechamber."

The molecular weight of the simulated PEG moiety is about 600 (degree of polymerization of 13). The experimental average molecular weight of mPEG2000-NHS which linked to the N terminal of peptide similar with other amino acid is 2000 (degree of polymerization of 45). We have verified the longer PEG (with degree of polymerization of 45) through a test simulation to compare with the used PEG (with degree of polymerization of 13) as shown in Figure as below. We found the longer PEG chain has no direct interaction with IR783 and peptide. In other words, the shorter PEG chain performs a similar behavior compared with the longer one. Therefore, the polymerization degree of 13 was used in the simulation process for the computation cost consideration. Take for instance, Wang et al. (Langmuir 2021, 37, 28, 8474–8485) also used atomistic MD simulations to reveal the interaction between PEG and protein. The degree of polymerization of PEG was 21 in the simulation, while PEG3k and PEG10k were

used in the corresponding experiments. These differences were reasonably accepted in the setting of PEG molecular weight.

The initial structure of amino acid sequence was relaxed starting from straight conformations for the sufficient conformation search.

We have added a simulation with a temperature of 310K. The result is shown in Figure as below. At the temperature setting of 300K and 310K respectively, it was clearly observed that the labeling of IR783 was almost perpendicular to the β -hairpin backbone and mPEG tail. When the mPEG motif was tailored, the IR783 alignment parallel to the backbone.

Moreover, we have corrected the misprint as “The timestep was 2 fs in production runs, and the coordinates were saved every 100 ps.”

2. At line 116, the citation to ref 22 seems to be inappropriate, because the cited paper is a review dealing with MD simulations of proteins at large pressures, so it is not particularly significant for the present study.

Response: Thanks for referee's suggestion. As the reviewer suggested we have removed this reference in the revised manuscript.

3. At lines 120-121 CASP12 is mentioned: why this notation? It should simply be aminoacid ASP12.

Response: Thank reviewer for pointing out the misleading we may cause, and we have modified the corresponding description as "The mPEG tail was close to the self-assembly motif through multiple hydrophobic interactions to stabilize its conformation, including hydrogen bonds VAL4:CYS8, ARG10:ALA1, ARG10:CY, ARG10:ASP12, ARG10:ASP12, and salt-bridge ARG10:ASP12 on the both side of the hairpin." All the revision has been highlighted in the revised manuscript.

4. What is the meaning of the coloured envelopes in the two left-most structures of Fig. 2? And anyway, are these the optimized geometry, or two snapshots taken during the MD run?

More importantly, the authors should provide some information about the molecular mobility (or rigidity) at room temperature or at physiological temperature to have a clue about the relevance of the reported β -hairpin conformations.

Response: Thanks for referee's suggestion. The colored envelopes in the two left-most structures of Fig. 2 shows the sum of the molecular interaction details including hydrogen bond, salt-bridge, and hydrophobic interaction. These figures were typical snapshots taken after the structure was stabilized. The radius of gyration (Rgyr) and root mean square calculation (RMSF) of AKLVFFGCGRGD sequences of M1 and R-M1 were calculated, and the results were shown in Supplementary Fig. 11. The initial structure of amino acid sequences was relaxed starting from straight conformations; therefore, we used the simulated trajectory of the last 100 ns to calculate RMSF. The results show that the conformational changes of M1 and R-M1 could be ignored, and both M1 and R-M1 can maintain β -hairpin conformations well.

Supplementary Fig. 11 | The radius of gyration (Rgyr) (a) and root mean square fluctuation (RMSF) (b) of AKLVFFGCGRGD sequences of M1 and R-M1.

5. At line 148, on the basis of circular dichroism spectra the authors find "well-ordered β -sheet assembled secondary structures", Indeed, these are somehow consistent with the MD simulations: in fact, at line 130 the author anticipate that the conformation of R-M1 would be "beneficial to the occurrence of intermolecular dynamic assembly". Simulation of this intermolecular assembly were not performed, but it is quite possible that the assembly (or

aggregation) may enhance the β -sheet ordering: could the authors briefly comment on this issue?

Response: Thanks a lot for the reviewer's comments. Through the MD simulation calculations, it was clearly observed that for M1, the IR783 was almost perpendicular to the β -hairpin backbone and mPEG tail, which preventing the further intermolecular assembly. When the mPEG motif was tailored, the IR783 was arranged parallel to the backbone, the hydrophilicity of molecule decreased and the hydrogen bonds on the self-assembled surface were exposed. The tailoring of the mPEG motif promotes the occurrence of intermolecular dynamic assembly, which is consistent with the experimental result.

I hope you will enjoy reading our revised manuscript. I look forward to your response and to that of the reviewers.

Sincerely,

Prof. Dr. Li-Li Li

CAS Key Laboratory for Biomedical Effects of Nanomaterials & Nanosafety
National Center for NanoScience and Technology (NCNST)

REVIEWERS' COMMENTS

Reviewer #1 (Remarks to the Author):

The authors have address my concern, and I support the acceptance of this work

Reviewer #2 (Remarks to the Author):

The research is comprehensive, and the results are significant for contrast agent/IR fluorescence probe design for pancreatic tumor imaging research and have clinical relevance.

I support the acceptance of this manuscript after the authors address the issues listed in the attached MS Word file.

Reviewer #3 (Remarks to the Author):

The authors have improved the manuscript. Some minor comments are listed below.

1. Fig. 5h and Supplementary Fig. 23: I do not think “transferred cells” is a great description. “Migrated cells” would better reflect the assay.
2. Fig. 6: Please clarify the location of the tumors in the figure legend.
3. Supplementary Fig. 18: Labels missing.
4. Supplementary Fig. 19-22: Controls for each experiment are missing. Add specific controls for each experiment (e.g., RGE for RGD) or at least show Fig. 19-22 together with Fig. 18 (choose one set from Fig. 18 unless they are from different experiments, in which case the panels need to be labeled properly).

Reviewer #4 (Remarks to the Author):

The authors have carefully considered all my comments and suggestions, and they responded satisfactorily by some minor changes to the manuscript. Accordingly, I feel that the manuscript can now be accepted for publication.

Title: "A bioactivated in vivo assembly (BIVA) nanotechnology fabricated NIR probe for small pancreatic tumor intraoperative navigation imaging" (Manuscript ID: NCOMMS-21-15472A)

Our point-by-point responses to reviewers:

Reviewer: 1

Comments: The authors have address my concern, and I support the acceptance of this work

Response: Thanks a lot for the reviewer's comments.

Reviewer: 2

Comments: The research is comprehensive, and the results are significant for contrast agent/IR fluorescence probe design for pancreatic tumor imaging research and have clinical relevance.

I support the acceptance of this manuscript after the authors address the issues listed in the attached MS Word file.

Thanks for addressing my previous questions. The following are the questions that were either unaddressed or not completely addressed. While the revision has made significant improvements, I suggest increasing language clarity.

1. Please add "and Supplementary Table 1" after "As shown in Fig. 3a".

Response: Thank for the advice of reviewer. We have revised the manuscript accordingly.

infrared (FTIR) spectroscopy, and wide-angle X-ray scattering (WAXS) spectroscopy were applied. As shown in Fig. 3a and Supplementary Table 1, CD spectrum of **M3** assemblies had a positive band at $\lambda=193$ nm and two negative bands at $\lambda=208$ nm, and $\lambda=225$ nm respectively, which implied a β -sheet and α -helix hybrid structure. In contrast, under the same concentration, **M1** molecules had a random coil secondary structure in CD spectrum as monomers which the concentration is lower than CAC. The FITR spectra of **M1** and **M3** in Fig. 3b

2. This sentence "M1 molecules had a random coil secondary structure in CD spectrum as monomers" is confusing. How can a monomer have a secondary structure?

Response: Sorry for the confusion. Since the molecular concentration used in the CD spectrum experiment of M1 molecules is 200 μ M, which is much lower than the CAC of M1 molecule (> 500 μ M). Therefore, we judge M1 molecules as monomer. It has been described in the manuscript.

infrared (FTIR) spectroscopy, and wide-angle X-ray scattering (WAXS) spectroscopy were applied. As shown in Fig. 3a and Supplementary Table 1, CD spectrum of **M3** assemblies had a positive band at $\lambda=193$ nm and two negative bands at $\lambda=208$ nm, and $\lambda=225$ nm respectively, which implied a β -sheet and α -helix hybrid structure. In contrast, under the same concentration, **M1** molecules had a random coil secondary structure in CD spectrum as monomers which the concentration is lower than CAC. The FITR spectra of **M1** and **M3** in Fig. 3b

3. Please change “tag” to “arrow heads” in this sentence: “The peaks at 1629 cm⁻¹, 1675 cm⁻¹, and 1698 cm⁻¹ of M3 indicated anti-parallel β-sheet structure (represent by green tag), peaks at 1648 cm⁻¹ and 1663 cm⁻¹ indicated parallel beta sheet structure (represent by blue tag)”

Response: Thank you for your suggestions. It has been described in the manuscript.

observed the intermolecular interactions. The peaks at 1629 cm⁻¹, 1675 cm⁻¹, and 1698 cm⁻¹ of M3 indicated anti-parallel β-sheet structure (represent by green arrow heads), peaks at 1648 cm⁻¹ and 1663 cm⁻¹ indicated parallel beta sheet structure (represent by blue arrow heads), the existence of 1654cm⁻¹ indicate α-Helix structure in M1 which verified the results deduced by CD in Fig 3a.^[28, 29] The evidence indicated that without

4. When I googled “Dundal phenomenon”, there was no hit. PLEASE ADD A SENTENCE TO EXPLAIN WHAT DUNDAL PHENOMENON IS AND CITE A REFERENCE in “the assemblies in aqueous solution had an obvious dundal phenomenon”.

Response: Thank you for your suggestions. I'm sorry that your reading is confused because of a writing error. We have modified "Dundal phenomenon" to " Tyndall phenomenon". It has been described in the manuscript.

3c). The R-M1 molecules had a rapid dynamic assembly (within few minutes), and the assemblies in aqueous solution had an obvious Tyndall phenomenon. As a homologous sequence with amyloid β-protein (Aβ), the self-assembly motif with peptide sequence of KLVFFGCG had similar aggregation kinetics to (Aβ)₄₂ peptide, which occurred via dynamic growth from oligomers to amyloid fibrils.^[30] The aggregation started from the

in DI water under a concentration of 200 μM. b, The FTIR spectra of M1 and M3, powder samples collected from freeze dried sample solutions. c, The typical β-sheet CD spectra of R-M1 in DI water (insert figure: the Tyndall phenomenon) under a

5. “Fig. 3 | Assembled structure conformations and self-assembly behavior in aqueous solution”, are ALL your samples were measured in aqueous solution or some were in the freeze-dried form like those for FT-IR measurements? Please be accurate.

Response: Thank you for your suggestions. The assembly samples used in Fig. 3 are all from aqueous solution. In order to facilitate the experiment, the freeze-dried samples were used in FT-IR measurements and WAXS spectrum. We have made supplementary explanations in the corresponding position of the manuscript.

Fig. 3 | Assembled structure conformations and self-assembly behavior in aqueous solution. a, CD spectra of M1 and M3 in DI water under a concentration of 200 μM. **b,** The FTIR spectra of M1 and M3, powder samples collected from freeze dried sample solutions. **c,** The typical β-sheet CD spectra of R-M1 in DI water (insert figure: the Tyndall phenomenon) under a concentration of 100 μM. **d,** The FTIR spectra of dynamic growth of R-M1, powder samples collected from freeze dried sample solutions in different period. **e,** Analysis of secondary structure composition of M1, M3, and R-M1 based on CD spectra. **f,** Elongation-nucleation-growth-procedure with ThT-staining. The mean of data of three samples with the same conditions is shown and data are presented as mean values +/-SD (n=3) **g,** The WAXS spectrum and illustration of the R-M1 fibrils, powder sample collected from freeze dried sample solution. **h,** The TEM images of nanofibers morphology of R-M1.

6. There are multiple issues in Fig. 4 legend. **“Fig. 4 | The FAP- α specific molecule tailoring and inducing *in situ* assembly. a,** The illustration the working mechanism of BIVA probe based on FAP- α catalytic hydrolysis. **b,** HPLC curves of **M3** and **M5** after incubation with FAP- α in buffer. **R-M3, M3** and **M5** were synthesized controls. The TEM image **c,** and the MALDI-TOF **d,** results of **M1** after tailoring by FAP- α in buffer. **e,** The enzyme specificity of **M1** in buffer. Buffer: 50 mm Tris, 1m NaCl, 1 mg/mL, BSA, pH 7.5; Concentration of **M1** and **M3**: 100 μ M; Concentration of FAP- α : 50 mM; Incubation time: 12 h.”

Fig. 4b legend is unclear. You have “R-M3 27.4 min” and “R-M3 27.6 min” as labels for the 1st figure of fig 4b, which one is which? In the second figure, you have two “M5 24.0 min”, are they different?

Fig. 4c legend is unclear. What are the red arrows pointing to? Are they M1+FAP- α ? Please add a concise illustration.

Fig. 4e, if you mean molar concentration, then “50 mm” needs to be written as “50 mM” and “1m” needs to be written as “1 M”.

Response: Thank you for your suggestions.

1. For easier identification, we use different colors for label and distinguish the two kind of **R-M3**. One **R-M3** is obtained by Co-incubation of **M3** and FAP- α , and the other is chemically synthesized. The same thing happened in **M5**. It has been described in the Fig.4c.

at 27.4 min can be identified by the R-M3 control (27.6 min). In the meantime, the wide peak at 14.6 min might be the remaining PEG residue. In sharp contrast, after incubation of FAP- α with M5, there was no change of the retention peak, which double confirmed that the GPA was the FAP- α specific recognized sequence and the molecule was cut between the amino acid of Pro and Ala. The retention time of M1 after co-incubation

2. We have made supplementary explanations in the corresponding position of the manuscript.

and **M5** were synthesized controls. The TEM image **c,** and the MALDI-TOF **d,** results of **M1** after tailoring by FAP- α in buffer, the red arrows represent the assembled fibrils of M1 after incubation with FAP- α . **e,** The enzyme specificity of **M1** in buffer.

3. We revised the corresponding position of the manuscript.

the red arrows represent the assembled fibrils of M1 after incubation with FAP- α . **e,** The enzyme specificity of **M1** in buffer. Buffer: 50 mM Tris, 1 M NaCl, 1 mg/mL, BSA, pH 7.5; Concentration of **M1** and **M3**: 100 μ M; Concentration of FAP- α : 50 μ M; incubation time: 12 h. The mean of data of four samples with the same conditions is shown and data are presented as mean values \pm SD (n = 4). $p = 2.04E-22 < 0.001$, p values were performed with one-way ANOVA by post-hoc Tukey's test for the indicated comparison.

7. “Fig. 5 | The BIVA effect with enhanced targeting to pancreatic tumor cell for outline location imaging.” What does “outline location imaging” mean? Do you mean for imaging tumor cell boundaries? Please rephrase.

Response: Sorry for the reading confusion caused by our vague description. what we want to express is “imaging tumor cell boundaries”. We revised the corresponding position of the manuscript.

Fig. 5 | **The BIVA effect with enhanced targeting to pancreatic tumor cell for boundaries imaging.** a, Illustration of BIVA

8. “In sharp contrast, after incubation of FAP- α with **M5**, there was no change of the retention peak, which double conformed that the GPA was the FAP- α specific recognized sequence and the molecule was cut between the amino acid of Pro and Ala.” What does “double conformed” mean? Conform with what? Do you actually mean “double confirmed”?

Response: Sorry for the reading confusion caused by our vague description. What we want to express is “double confirmed”. We revised the corresponding position of the manuscript.

at 27.4 min can be identified by the **R-M3** control (27.6 min). In the meantime, the wide peak at 14.6 min might be the remaining PEG residue. In sharp contrast, after incubation of FAP- α with **M5**, there was no change of the retention peak, which double confirmed that the GPA was the FAP- α specific recognized sequence and the molecule was cut between the amino acid of Pro and Ala. The retention time of **M1** after co-incubation

9. For Fig. 6c, was the background signal subtracted? how was it subtracted? Please clarify this in method section.

Response: Thank you for your suggestions. AUC data comes from the time-dependent quantitative calculation of the average fluorescence intensity in tumor area. We have made supplementary explanations in the corresponding position of the manuscript.

circulation-half-life. c, The time-dependent quantitative calculation of the average fluorescence intensity in tumor area and the area under the curve (AUC) of ICG, **M2** and **M1**. The mean of three biological replicates is shown and data are presented as mean values \pm SD (n = 3).

10. “In order to further understand the contribution of the active targeting and AIR effect during locating of the cells, the cell transwell experiment was used to quantitatively evaluate interference of the cell migration based on different molecules (Fig. 5f)”

It is unclear what “based on different molecules” means. What different molecules? Please clarify by re-phrasing it.

Response: Thank you for your suggestions. In Transwell experiment, we used molecules **M1**, **M2** and **M3** as control. We have made supplementary explanations in the corresponding position of the manuscript.

In order to further understand the contribution of the active targeting and AIR effect during locating of the cells, the cell transwell experiment was used to quantitatively evaluate interference of the cell migration based on M1, M2, and M4 (Fig. 5f). As expected, untreated cells were easy to migrate to the lower chamber, while

11. In fig. 8 legend, please clarify what the error bars stand for, are they standard errors of the means (SEM) or standard deviations (SD)?

Response: Thank you for your suggestions. The error bars in fig. 8 are standard errors of standard deviations (SD). We have made supplementary explanations in the corresponding position of the manuscript.

Fig.8 | Acute toxicity evaluation. a, Liver function indicators: alanine aminotransferase (ALT), alkaline phosphatase (ALP), aspartate aminotransferase (AST), total protein (TP) and albumin concentration (ALB). The mean of four biological replicates is shown and data are presented as mean values \pm SD (n=4). b, Blood biochemical indicators: creatinine (CREA), white blood cells (WBC), red blood cells (RBC), hematocrit (HCT), mean corpuscular volume (WCV), mean corpuscular haemoglobin concentration (MCH), mean corpuscular hemoglobin concentration (MCHC), red cell volume distribution width-coefficient of variation (RDW-CV), platelet (PLT) and mean platelet volume (MPV). The mean of four biological replicates is shown and data are presented as mean values \pm SD (n=4). c, Histologic sections of different organs: heart, liver, spleen, lung, kidney and pancreas compared with healthy group (PBS). Staining: H&E. Injection dose of M1 (i.v. administration): 16 mg/kg. Administration time: 24 h. Scale bar 200 μ m.

Under Methods section:

12. **Cellular imaging experiment:** Please add optical channel information, i.e. Excitation and emission wavelength or band.

Response: Thank you for your suggestions. We have made supplementary explanations in the corresponding position of the manuscript.

Cellular imaging experiment: MIA PaCa-2 cells were digested with trypsin, counted with cell counting plate, diluted to 1×10^4 with DMEM medium, and 1 mL to 1 cm diameter confocal microscope dish was taken. The cells were incubated in 37% CO₂ atmosphere for 24 hours. The medium containing M1 and M2 (50 μ M) was replaced and incubated with the cells for 2 hours. The cells were washed with PBS for 3 times. CLSM (Zeiss 710) was used to image cells under 40 \times objective, excitation = 488 nm, and emission > 520 nm.⁴

13. Why only female mice were tested with?

Response: Thank you for your question. There is no evidence that the incidence of pancreatic cancer is related to gender, so there is no specific gender requirement *in vivo*. Compared with male mice, female mice are more docile and suitable for group feeding. Male mice are easy to fight, their physical condition is unstable, and are easy to produce experimental errors. Therefore, we selected female mice as experimental subjects.

14. **“Tumor slices and staining:** Tumors were harvested and collected in 4 % paraformaldehyde solution after treated with M1 at a dose of 16 mg/kg for 12 h and 48 h.”

Do you mean “tumors were harvested after treated with M1 at a dose of 16 mg/kg for 12 h then fixed in 4 % paraformaldehyde solution for 48 h”? Please clarify.

Response: Sorry for the reading confusion caused by our vague description. The tumors were sectioned 12 h and 48 h after injection of fluorescent probe. We revised the corresponding position of the manuscript.

Tumor slices and staining: Tumors were harvested and collected in 4 % paraformaldehyde solution after tumor-bearing mice treated with M1 at a dose of 16 mg/kg for 12 h and 48 h. The Hematoxylin and Congo red staining procedure was performed by Google biotechnology (Wuhan) Co. LTD.

15. Please explain what x and y axis labels stand for in the figure legends for both Supplementary Fig. S9 and Supplementary Fig. S10.

Response: Thank you for your question. We have made supplementary explanations in the corresponding position of the Supplementary.

Supplementary Fig. 9 | The critical aggregation concentration (CAC) of M3 and M5 was measured by using pyrene as a probe. Experiments were repeated three times. The x axis is logarithm concentration and the y axis is the ratio of peak intensities of 374 nm and 383 nm. E_{374}/E_{383} .

Reviewer: 3

Comments: The authors have improved the manuscript. Some minor comments are listed below.

1. Fig. 5h and Supplementary Fig. 23: I do not think “transferred cells” is a great description. “Migrated cells” would better reflect the assay.

Response: Thank reviewer for reminding. We had changed the description in manuscript according reviewer’s advice.

infrared (FTIR) spectroscopy, and wide-angle X-ray scattering (WAXS) spectroscopy were applied. As shown in Fig. 3a and Supplementary Table 1, CD spectrum of M3 assemblies had a positive band at $\lambda=193$ nm and two negative bands at $\lambda=208$ nm, and $\lambda=225$ nm respectively, which implied a β -sheet and α -helix hybrid structure.

2. Fig. 6: Please clarify the location of the tumors in the figure legend.

Response: We are sorry for confusing. We had added description in figure legend as “The circles represent the locations of the tumors”, as shown below:

Fig. 6 | The BIVA effect optimized the metabolism of the probe *in vivo*. **a**, The time-dependent NIR fluorescence image of mice bearing MIA PaCa-2 cells after intravenous administration of ICG, M2 and M1 with a dose of 16 mg/kg. The images acquired at time intervals from 0.1 h to 120 h are managed with the same procedure. (The circles represent the locations of the tumors). **b**, the blood circulation curve of ICG, M2 and M1 based on exponential curve fitting. The $t_{1/2}$ value was the blood

3. Supplementary Fig. 18: Labels missing.

Response: Sorry for the confusing. And we added labels and description in figure legend for this three independ representative experiment , as shown below:

■ Supplementary Fig. 18 | Representative confocal images in three independent experiment of **M1** with BIVA effect on Miaapaca-2 cells after incubation for 2 h.

4. Supplementary Fig. 19-22: Controls for each experiment are missing. Add specific controls for each experiment (e.g., RGE for RGD) or at least show Fig. 19-22 together with Fig. 18 (choose one set from Fig. 18 unless they are from different experiments, in which case the panels need to be labeled properly).

Response: We are sorry for the data display. All the Supplementary Fig. 19-22 were control experiments for Fig. 5c to provide BIVA effect of **M1**, we showed the Fig. 19-22 together with Fig. 5c in updated Supplementary Fig. 19 for better reading.

■ **Supplementary Fig. 19** | CLSM images of Miapaca-2 cells treated under different conditions. **a**, CLSM images of **M1** with BIVA effect on Miapaca-2 cells after incubation for 1 h. **b**, **M1** incubated αv integrins blocked Miapaca-2 cells for 1 h and washed by fresh DMEM for three times (Miapaca-2 cells were treated by RGD for 1h at 50 μM and washed by fresh DMEM for three times before adding **M1**). **c**, **M1** incubated Fap- α inhibited Miapaca-2 cells for 1h and washed by fresh DMEM for three times (Miapaca-2 cells were treated by Fap- α inhibitor Ac-Gly-BoroPro for 1h at 50 nM and washed by fresh DMEM for three times before adding **M1**). **d**, **R-M1** incubated Miapaca-2 cells after incubation for 1 h without washing. **e**, **M4** incubated Miapaca-2 cells after incubation for 1 h and washed by fresh DMEM for three times.

Reviewer: 4

Comments: The authors have carefully considered all my comments and suggestions, and they responded satisfactorily by some minor changes to the manuscript. Accordingly, I feel that the manuscript can now be accepted for publication.

Response: Thanks a lot for the reviewer's comments.

I hope you will enjoy reading our revised manuscript. I look forward to your response and to that of the reviewers.

Sincerely,

Prof. Dr. Li-Li Li

CAS Key Laboratory for Biomedical Effects of Nanomaterials & Nanosafety
National Center for NanoScience and Technology (NCNST)